# A decision point between transdifferentiation and programmed cell death priming controls KRAS-dependent pancreatic cancer development

KRAS-dependent acinar-to-ductal metaplasia (ADM) is a fundamental step in the development of pancreatic ductal adenocarcinoma (PDAC), but the involvement of cell death pathways remains unclear. Here, we show that key regulators of programmed cell death (PCD) become upregulated during KRAS-driven ADM, thereby priming transdifferentiated cells to death. Using transgenic mice and primary cell and organoid cultures, we show that transforming growth factor (TGF)-β-activated kinase 1 (TAK1), a kinase regulating cell survival and inflammatory pathways, prevents the elimination of transdifferentiated cells through receptor-interacting protein kinase 1 (RIPK1)-mediated apoptosis and necroptosis, enabling PDAC development. Accordingly, pharmacological inhibition of TAK1 induces PCD in patient-derived PDAC organoids. Importantly, cell death induction via TAK1 inhibition does not appear to elicit an overt injury-associated inflammatory response. Collectively, these findings suggest that TAK1 supports cellular plasticity by suppressing spontaneous PCD activation during ADM, representing a promising pharmacological target for the prevention and treatment of PDAC.

Pancreatic ductal adenocarcinoma (PDAC) has remained one of the most challenging cancers to treat, as even novel therapeutic approaches, such as immunotherapy or personalized treatment methods, have yielded disappointing results. Therefore, a deeper understanding of the underlying processes that occur during cancer initiation and establishment is critical and might allow the development of chemopreventive strategies in high-risk PDAC clinical settings, such as chronic pancreatitis or genetic predisposition[1,2].

During cancer progression, tumor cells can undergo molecular and phenotypic changes collectively referred to as cellular plasticity. For example, chronic pancreatitis or oncogenic mutations in Kirsten rat sarcoma viral oncogene homolog (KRAS), which are found in most PDACs[3], can induce acinar-to-ductal metaplasia (ADM), a reprogramming process that leads to the dedifferentiation of pancreatic acinar cells and their transdifferentiation into progenitor-like cells with ductal

characteristics[4]. While ADM facilitates pancreatic regeneration after injury, persistent ADM, especially in the presence of oncogenic *KRAS* mutations, increases the potential of transdifferentiated cells to progress into pancreatic intraepithelial neoplasia (PanIN) and ultimately PDAC[5,6]. ADM is a highly plastic process involving extensive transcriptional rewiring leading to reduced expression of acinar-specific genes, such as Mist, amylase, carboxypeptidase, and elastase, and increased expression of ductal-specific genes, such as cytokeratin-19 (CK-19), cytokeratin-20 (CK-20), SOX9, and carbonic anhydrase[7]. However, it is unclear whether these early events of pancreatic carcinogenesis are facilitated by specific pro-survival molecules, which could be amenable to pharmacological targeting during ADM and PDAC development.

The multifunctional kinase transforming growth factor (TGF)-β-activated kinase 1 (TAK1) has recently emerged as a signaling node with

✉e-mail: luedde@hhu.de

diverse and tumor type-specific functions in cancer initiation and progression[8]. TAK1 is activated downstream of tumor necrosis factor (TNF) and TGF-β, two cytokines known to contribute to ADM[6,9]. It mediates cell survival not only through phosphorylation of the IκB kinase (IKK) complex catalytic subunits IKKα/IKK1 and IKKβ/IKK2 and subsequent activation of nuclear factor kappa B (NF-κB)[10], but also through suppression of programmed cell death (PCD) by mediating inhibitory phosphorylation of receptor-interacting serine/threonine protein kinase 1 (RIPK1), a master regulator of CASPASE 8 (CASP8)-dependent apoptosis and RIPK3-dependent necroptosis[11–14].

In this work, we examined the role of TAK1 in KRAS-dependent PDAC development. We could show that TAK1 determines whether acinar cells will survive, transdifferentiate into ductal cells and subsequently develop into cancer cells or will succumb to PCD through RIPK1-dependent necroptosis and apoptosis. Consequently, genetic deletion or pharmacological inhibition of TAK1 not only prevented the development of PanIN and PDAC in vivo and in vitro independently of its function in NF-κB activation, but also induced death of established patient-derived pancreatic tumor spheroids. Collectively, our findings suggest that PCD induction by targeting TAK1 may represent a clinically translatable prevention and treatment strategy for PDAC.

## Results

### TAK1 deficiency suppresses KRAS-driven ADM and PDAC development

In order to assess the expression levels and activation status of the prosurvival kinase TAK1 in PDAC, we first performed immunohistochemical (IHC) staining in a tissue microarray of 173 human PDAC samples (Supplementary Table 1). In this analysis, we detected a strong upregulation and activation of TAK1 in cancer cells (Fig. 1a, b; Supplementary Fig. 1a, b). In line, the TAK1-binding protein 3 (TAB3), which is essential for TAK1 activation[15], was also found to be highly expressed in PDAC samples (Supplementary Fig. 1c, d). To further examine the role of TAK1-dependent signaling in vivo, we intercrossed transgenic mice with a floxed, constitutively active *Kras* allele (KRAS$^{G12D-Fl/+}$ mice)[16] and *Tak1*$^{Fl/Fl}$ mice[17] with *Ptf1a*-cre[18] mice to generate animals with either oncogenic KRAS expression in acinar cells alone (KRAS$^{G12D}$) or mice with combined acinar cell (Ac)-specific oncogenic KRAS expression and deletion of *Tak1* (KRAS$^{G12D}$ TAK1$^{ΔAc}$). Analyses of these animals at 6 weeks of age showed no gross abnormalities in all groups (Supplementary Fig. 2a). As expected, KRAS$^{G12D}$ and KRAS$^{G12D}$ TAK1$^{ΔAc}$ mice showed similar levels of constitutive activation of KRAS (Supplementary Fig. 2b). Moreover, acinar cells from KRAS$^{G12D}$ TAK1$^{ΔAc}$ mice displayed an impairment of IκBα degradation upon TNF stimulation, confirming the NF-κB signaling inhibition due to the lack of the upstream kinase TAK1 (Supplementary Fig. 2c, d).

Histological analyses of the pancreas in 18-week-old KRAS$^{G12D}$ mice revealed the presence of ADM areas and PanIN-1 and PanIN-2 lesions as potential precursors of PDAC[7] (Fig. 1c, d). Strikingly, this process was abolished in KRAS$^{G12D}$ TAK1$^{ΔAc}$ mice (Fig. 1c, d), suggesting that TAK1-dependent signaling plays a crucial role during ADM. Of note, analyses of single TAK1$^{ΔAc}$ mice (without KRAS activation) did not reveal any microscopically detectable pancreatic abnormalities at the age of 18 weeks (Supplementary Fig. 2e, f), indicating that TAK1 deficiency does not significantly affect normal acinar cell development but specifically influences the process of KRAS-driven ADM. To further substantiate this inhibitory effect of *Tak1* deletion, we implemented a three-dimensional (3D) collagen matrix culture system of pancreatic acinar cell explants (Fig. 1e). In this 3D culture system that mimics the observed ADM process in the pancreas, acinar cells isolated from 6-week-old mice KRAS$^{G12D}$ mice were let to transdifferentiate into ductal-like cells and form duct-like structures within the collagen matrix in the absence of epidermal growth factor (EGF). Under these culture conditions, KRAS$^{G12D}$-expressing acinar cells could efficiently undergo

ADM and form duct-like structures, as assessed by bright field microscopy, H&E staining and IHC for the ductal cell marker SOX9[19], while wild-type (WT) cells did not form duct-like structures (Fig. 1f, g). Interestingly, KRAS-activated acinar cells failed to form ADM structures upon concomitant TAK1 deficiency (Fig. 1f, g). We also tested if pharmacological TAK1 inhibition had a similar effect on KRAS$^{G12D}$-expressing acinar cells using the TAK1 kinase activity inhibitor 5Z-7-Oxozeaenol[20]. Indeed, we observed that the TAK1 inhibitor prevented duct-like structure formation in a concentration dependent manner (Fig. 1h–j; Supplementary Fig. 2g), indicating that the function of TAK1 in this process was kinase activity-dependent. In contrast, the use of the IKK inhibitor TPCA-1 in a concentration that sufficiently suppressed the expression of NF-κB target genes (Supplementary Fig. 2h, i) did not block the formation of duct-like structures (Fig. 1i, j), suggesting that ADM impairment upon TAK1 deficiency was not due to NF-κB inhibition.

To further evaluate the effect of TAK1 deficiency on carcinogenesis, we monitored the mice with different genotypes over a period of 30 and 52 weeks. In KRAS$^{G12D}$ mice, ADM and pre-neoplastic lesions progressed to high-grade lesions (PanIN-3) and advanced PDAC (Fig. 1k, l; Supplementary Fig. 2j, k). In stark contrast, aged KRAS$^{G12D}$ TAK1$^{ΔAc}$ mice were largely devoid of PanIN-3 lesions and PDAC (Fig. 1k, l; Supplementary Fig. 2j, k). Finally, as TPCA, in contrast to 5Z-7-Oxozeaenol, did not block ADM in vitro, we assessed the effect of NF-κB inhibition in vivo by deleting the NF-κB subunit *Rela* in KRAS-driven PDAC-development[21]. Contrary to the tumor-preventing effect of *Tak1* ablation in KRAS-driven PDAC development, ablation of *Rela* did not prevent ADM and PDAC formation in KRAS$^{G12D}$ mice (KRAS$^{G12D}$ RelA$^{ΔAc}$), but instead, it even enhanced tumorigenesis (Supplementary Fig. 2l, m). Taken together, our findings showed that TAK1 ablation blocked KRAS-driven ADM/PanIN formation and PDAC development in an NF-κB independent manner.

### Activation of TAK1-regulated PCD pathways prevent ADM

To assess the molecular mechanism how TAK1 prevented KRAS-driven ADM in the pancreas, we first performed immunoblotting analyses to examine the activation of pathways associated with MAP kinase signaling and proliferation in pancreatic protein lysates from 6- and 18-weeks-old mice. In line with our finding that pancreatic KRAS activation was unaffected by *Tak1* deletion in 6-week-old mice (see Supplementary Fig. 2b), we did not detect clear differences in the phosphorylation of downstream targets of KRAS signaling, such as protein kinase B (AKT), extracellular signal-regulated kinase (ERK) and mitogen-activated protein kinase kinase 1/2 (MEK1/2) in pancreatic lysates of 6-week-old WT, TAK1$^{ΔAc}$, KRAS$^{G12D}$ and KRAS$^{G12D}$ TAK1$^{ΔAc}$ mice (Supplementary Fig. 3a). Conversely, analysis of whole pancreas lysates from 18-week-old mice revealed an increase in AKT, ERK and MEK1/2 phosphorylation in KRAS$^{G12D}$ mice, which was abolished upon additional deletion of *Tak1* (Supplementary Fig. 3b). Accordingly, IHC analysis of p-ERK in pancreatic sections from 18-week-old mice revealed a strong signal in ADM and PanIN lesions observed in KRAS$^{G12D}$ mice (Supplementary Fig. 3c). In contrast, the activation of the stress kinases mitogen-activated protein kinase 8 (JNK1), 9 (JNK2) and 14 (p38) was not affected in all mice at both time points (Supplementary Fig. 3a, b). Furthermore, evaluation of proliferation by Ki-67 immunolabelling revealed no differences in proliferation rate at 6 weeks of age, but high cell proliferation in areas of ADM/PanIN formation in 18-week-old KRAS$^{G12D}$ mice (Supplementary Fig. 3d–g). Analysis of the pancreatic acinar cell explants cultured on collagen matrices also showed proliferating cells mainly in the duct-like structures originating from KRAS$^{G12D}$-expressing acinar cells, while proliferating cells were not detected in explants from KRAS$^{G12D}$ TAK1$^{ΔAc}$ mice (Supplementary Fig. 3h). Together, these findings suggested that ERK and MEK1/2 signaling activation, as well as the increased ductal cell proliferation, reflected ADM/PanIN formation in

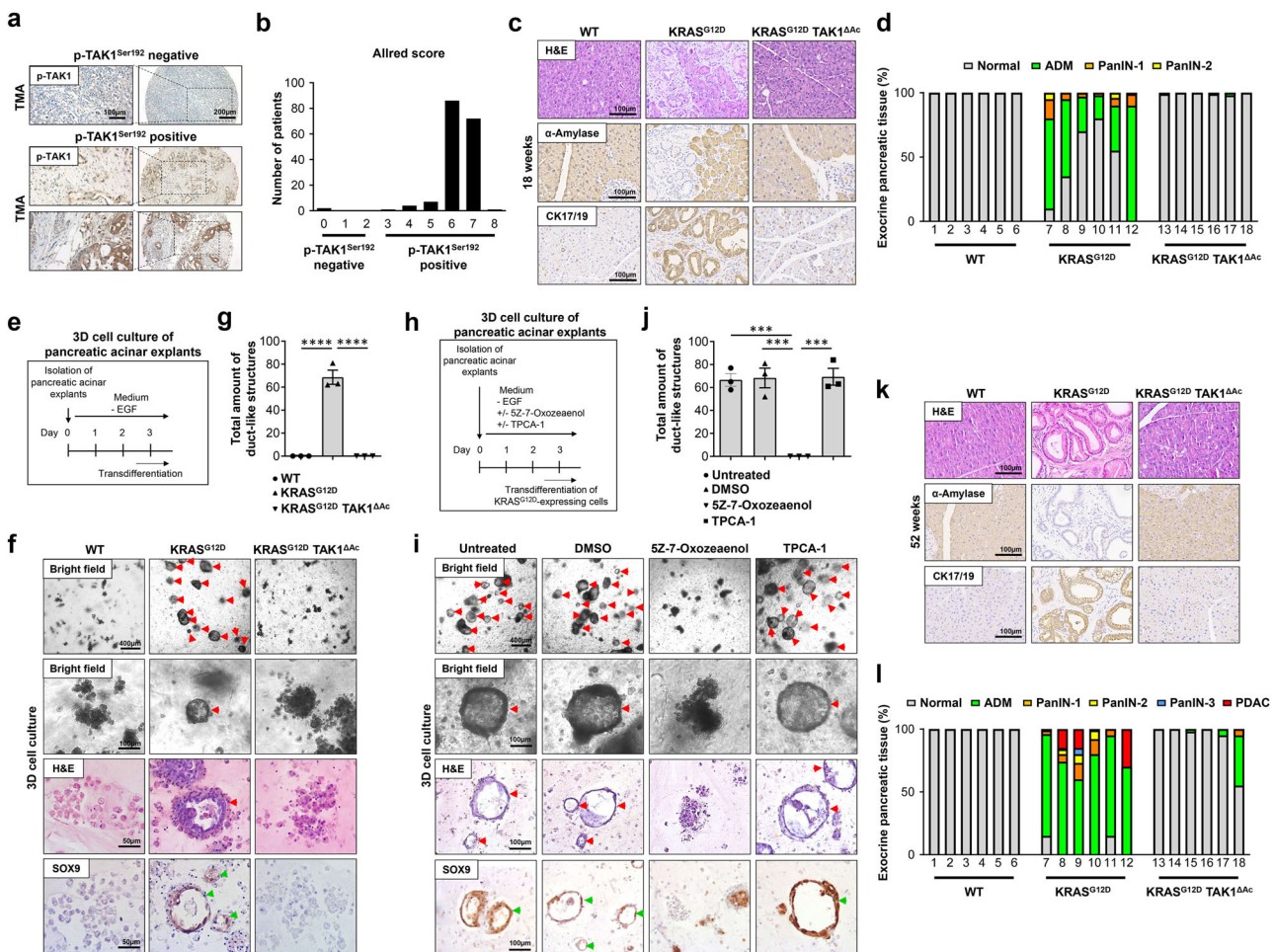

**Fig. 1 | TAK1 supports KRAS$^{G12D}$-driven ADM and PanIN formation resulting in PDAC development. a, b** Immunohistochemistry (IHC) of phospho-TAK1$^{Ser192}$ (p-TAK1 $^{Ser192}$) in samples of a human PDAC tissue microarray (TMA) and quantification using the Allred scoring system (Scores 0-2 = negative, Scores 3-8 = positive). $n = 173$, biologically-independent samples. **c, d** Representative images after H&E staining, α-Amylase and cytokeratin 17/19 (CK17/19) IHC on pancreatic tissue sections from 18-week-old mice and quantification of healthy pancreas tissue (Normal), acinar-to-ductal metaplasia (ADM), pancreatic intraepithelial neoplasia 1 (PanIN-1) and PanIN-2 in these animals ($n = 6$ mice per genotype). **e–g** Experimental design to study the transdifferentiation capability of pancreatic acinar cell explants in 3D collagen matrices. Representative images by bright-field microscopy, after H&E staining and SOX9 IHC. Duct-like hollow structures formed at day 3 are highlighted with red arrowheads (quantified in g). Green arrowheads indicate SOX9$^+$ duct-like structures. Results are expressed as mean ± SEM. The experiment was done with acinar explants from 3 different mice per genotype. $P$ value was calculated by ordinary one-way ANOVA (two-tailed) with Tukey's multiple-

comparisons test. ****$p < 0.0001$. **h–j** Experimental setting to study the transdifferentiation capability of KRAS$^{G12D}$-expressing pancreatic acinar cell explants in 3D collagen matrices in the presence of various inhibitors. Representative images by bright-field microscopy, after H&E staining and SOX9 IHC of KRAS$^{G12D}$-expressing pancreatic acinar explants grown in normal medium (untreated), or treated with DMSO, 5Z-7-Oxozeaenol (10 μM) or TPCA-1 (10 μM). Duct-like structures formed at day 3 are highlighted with red arrowheads and their total amount per genotype is indicated. Green arrowheads indicate SOX9$^+$ duct-like structures. Results are expressed as mean ± SEM. The experiment was done with acinar explants from 3 different mice per genotype. $P$ value was calculated by ordinary one-way ANOVA (two-tailed) with Tukey's multiple-comparisons test. ***$p = 0.0003$ untreated vs Oxo, ***$p = 0.0003$ DMSO vs Oxo, ***$p = 0.0002$ Oxo vs TPCA-1. **k, l** H&E staining, IHC of α-Amylase and CK17/19 on pancreatic tissue sections from the indicated 52-week-old mice and quantification of the different stages of pancreatic cancer development ($n = 6$ mice per genotype). Source data are provided in the Source Data file.

aging mice and argued against a direct regulation of these disease markers by TAK1.

We next investigated whether *Tak1* deletion had induced spontaneous PCD during the KRAS-driven ADM and carcinogenesis process in the pancreas. To this end, we analyzed the level of spontaneous apoptotic cell death in WT, KRAS$^{G12D}$, and KRAS$^{G12D}$ TAK1$^{ΔAc}$ mice at 6 and 18 weeks of age, but could not detect major differences between the different genotypes (Supplementary Fig. 3i, j). However, in a setting of slow progression of ADM formation in KRAS$^{G12D}$-expressing pancreatic tissue over several weeks, it might be difficult to detect spontaneous cell death as a single-cell event in TAK1-deficient pancreatic cells. Thus, we decided to explore the effect of TAK1 inhibition on PCD under more controlled conditions in vitro. To analyze cell death induction, we first cultured acinar cells isolated from WT,

KRAS$^{G12D}$ and KRAS$^{G12D}$ TAK1$^{ΔAc}$ mice in the presence of EGF, in order to trigger acinar-to-ductal transdifferentiation in all genotypes independently of KRAS$^{G12D}$ expression[22], and subsequently, we treated the transdifferentiated cells with TNF in the absence of EGF (Fig. 2a). Our immunoblotting analysis revealed CASPASE 3 (CASP3) cleavage, an apoptosis induction marker, and phosphorylation of mixed lineage kinase like (MLKL), a necroptosis induction marker, only in cells isolated from KRAS$^{G12D}$ TAK1$^{ΔAc}$ mice (Fig. 2b).

Next, we cultured KRAS$^{G12D}$-expressing pancreatic acinar explants on a 3D collagen matrix and allowed them to undergo ADM and form duct-like structures before treating them with 5Z-7-Oxozeaenol or DMSO (solvent) for 24 h (Fig. 2c–e). As expected, cells treated with DMSO showed SOX9$^+$ duct-like structures containing proliferating cells. In contrast, 5Z-7-Oxozeaenol treatment led

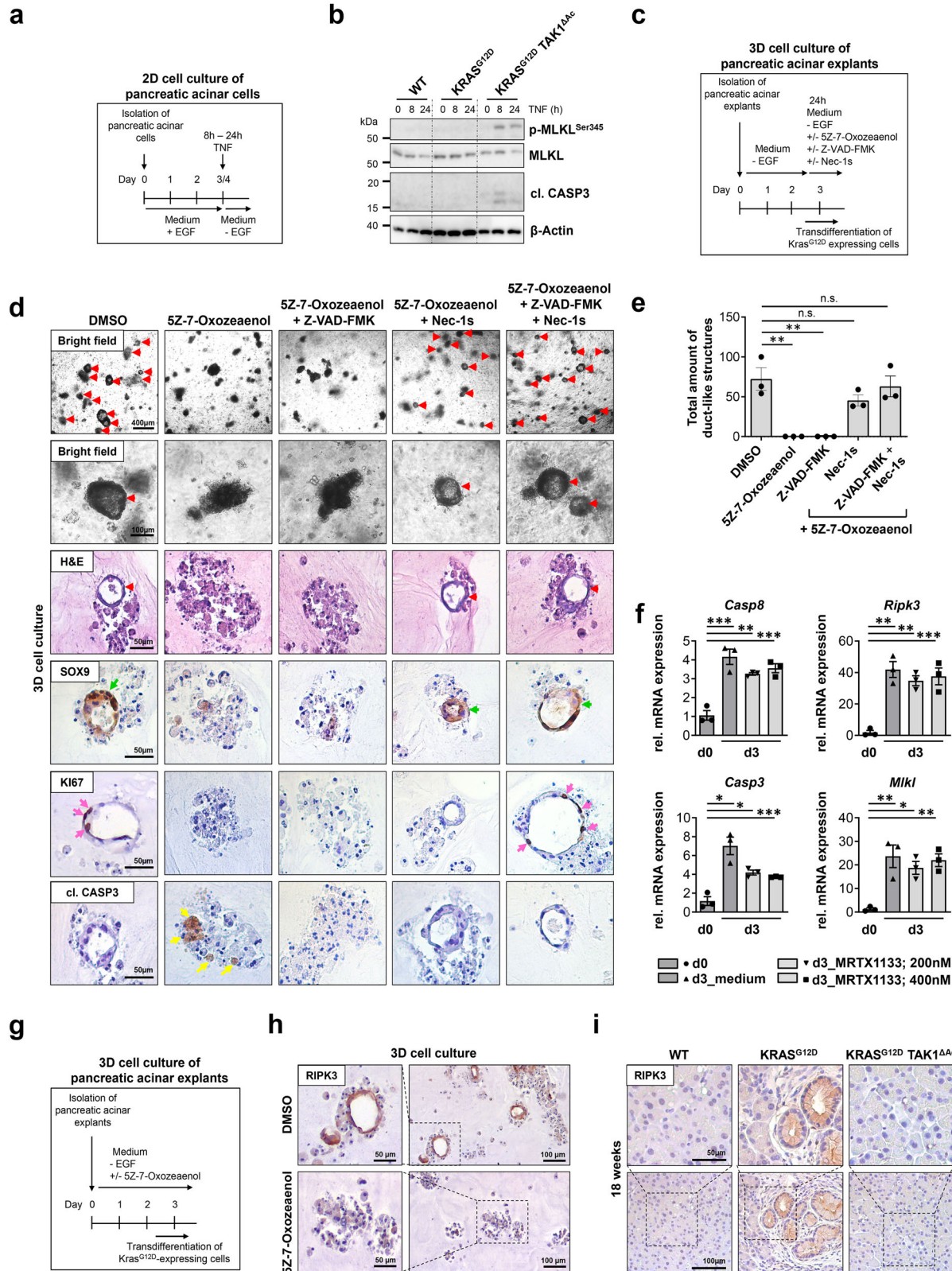

to the collapse of the duct-like structures that showed cleaved CASP3+ apoptotic cells, a feature that was not observed upon DMSO treatment alone (Fig. 2d). To further characterize the nature of PCD upon TAK1 inhibition, we treated duct-like structures established from acinar cell explants of KRAS$^{G12D}$ mice with 5Z-7-Oxozeaenol in combination with Z-VAD-*fmk* (pan-CASPASE inhibitor) and/or Nec-1s (inhibitor of RIPK1-dependent apoptosis and necroptosis) (Fig. 2c–e).

Interestingly, single Z-VAD-*fmk* treatment could not prevent the disintegration of ADM structures caused by TAK1 inhibition, suggesting that blockage of apoptotic cell death alone was not sufficient to prevent PCD induced upon treatment with the TAK1 inhibitor. On the contrary, the duct-like structures were preserved upon treatment with Nec-1s alone or in combination with Z-VAD-*fmk* (Fig. 2d, e). These data suggested that TAK1 inhibition sensitized cells

**Fig. 2 | Transdifferentiation sensitizes acinar cells to apoptotic and necroptotic PCD. a, b** Experimental design to study cell death induction in primary pancreatic acinar cells grown in 2D cell culture upon treatment with TNF (100 ng/ml). Immunoblotting analysis in lysates of pancreatic acinar cells isolated from the indicated mice. The experiment was done twice with one mouse per genotype. **c–e** Experimental design to examine the susceptibility of KRAS$^{G12D}$-expressing pancreatic acinar cell explants grown in 3D collagen matrices to PCD after transdifferentiation and formation of duct-like structures. The effect of DMSO, 5Z-7-Oxozeaenol (10 μM) alone or in combination with Z-VAD-*fmk* (25 μM), Nec-1s (50 μM) or both for 24 h was assessed. Representative images by bright-field microscopy, H&E staining and SOX9, Ki67 and cl. CASP3 IHC. Arrowheads highlight intact duct-like structures (red; quantified in e), SOX9$^+$ duct-like structures (green), Ki67$^+$ (pink) and cl. CASP3$^+$ (yellow) cells. Results are expressed as mean ± SEM. The experiment was done with acinar explants from 3 different KRAS$^{G12D}$ mice. *P* value was calculated by ordinary one-way ANOVA (two-tailed) with Tukey's multiple-comparisons test. **\*\*$p$ = 0.0016**,

n.s. = not significant. **f** qRT-PCR analysis of the mRNA expression of key apoptotic and necroptotic cell death mediators in 3D collagen matrices of KRAS$^{G12D}$ pancreatic acinar cell explants before (d0) and after (d3) transdifferentiation, treated with or without MRTX1133 (200 nM or 400 nM). The experimental design is shown in Supplementary Fig. 4a. All values were normalized to *Sdha* expression. Results are expressed as mean ± SEM. The experiment was done using acinar explants from the same three KRAS$^{G12D}$ mice. *P* values were calculated by ordinary one-way ANOVA (two-tailed) with Tukey's multiple-comparisons test indicated in the figure. **g, h** Experimental design to examine the upregulation of RIPK3 after transdifferentiation of KRAS$^{G12D}$-expressing pancreatic acinar cell explants in 3D collagen matrices upon DMSO or 5Z-7-Oxozeaenol (10 μM) treatment. Representative images after RIPK3 IHC. The experiment was done with acinar explants from 3 different KRAS$^{G12D}$ mice. **i** Representative images after RIPK3 IHC on pancreatic tissue sections from the indicated 18-week-old mice (*n* = 4 mice per genotype). Source data are provided in the Source Data file.

---

undergoing acinar-to-ductal transdifferentiation to RIPK1 kinase activity-dependent apoptosis and necroptosis.

To explore why KRAS$^{G12D}$-expressing acinar cells were sensitive to PCD during ADM in the absence of TAK1, we examined the expression of key mediators of apoptosis and necroptosis in KRAS$^{G12D}$-expressing acinar cells before and after their 3-day transdifferentiation into duct-like structures by qRT-PCR (Supplementary Fig. 4a). As expected, the expression of the acinar cell markers *Amylase 1 (Amy1)* and *Mist 1* was lost during ADM, while the progenitor and ductal cell marker *Cytokeratin 19 (Ck19)* was strongly upregulated (Supplementary Fig. 4b). Strikingly, we found that the expression of two essential necroptosis mediators, *Ripk3* and *Mlkl*, as well as two key apoptosis mediators, *Caspase 8 (Casp8)* and *Casp3*, was significantly upregulated upon acinar-to-ductal transdifferentiation (Fig. 2f). In contrast, the expression of other cell death-associated molecules, such as *Ripk1* and *Casp9*, was not significantly altered during the transdifferentiation process (Supplementary Fig. 4b). To confirm these data on protein level, we performed IHC staining for RIPK3 in samples of collagen matrix-cultured acinar explants. Indeed, we could detect a strong RIPK3 expression in the KRAS$^{G12D}$-expressing ductal cells that formed the ADM structures, which was not observed in 5Z-7-Oxozeaenol-treated acinar cells that failed to transdifferentiate (Fig. 2g, h). Accordingly, we could detect an increased RIPK3 expression in pancreatic ADM/PanIN lesions of KRAS$^{G12D}$ mice (Fig. 2i). Finally, to assess whether the observed upregulation of the PCD mediators is induced by mutant KRAS, we treated our acinar explants during our 3-day in vitro ADM assay with the recently developed, KRAS$^{G12D}$-specific inhibitor MRTX1133 that has shown anti-tumor efficacy in several PDAC mouse models[23–25]. As anticipated, MRTX1133 significantly inhibited the formation of duct-like structures in a dose-dependent manner (Supplementary Fig. 4c, d) without inducing, however, significant cytotoxicity (Supplementary Fig. 4e). Interestingly, our qRT-PCR analysis revealed that MRTX1133 neither reduced the upregulated expression of the PCD mediators, nor prevented the loss of acinar cell marker and induction of ductal cell marker expression (Fig. 2f; Supplementary Fig. 4b). To further test whether KRAS$^{G12D}$ expression is required to confer increased sensitivity to PCD, we expressed GFP or KRAS$^{G12D}$ in immortalized human pancreatic ductal epithelial cells (HPDEs) using a Doxycycline (Dox)-inducible system. Dox-mediated KRAS$^{G12D}$ induction for 3 days led to the development of characteristic cell vacuolization, indicative of increased liquid-phase endocytosis[26], and the upregulation of phosphorylated ERK (Supplementary Fig. 4f, g). Subsequent incubation of HPDEs with 5Z-7-Oxozeaenol alone or in combination with TNF led to a significant induction of cell death independent of KRAS$^{G12D}$ expression (Supplementary Fig. 4h). Altogether, these findings suggest that important apoptosis and necroptosis mediators are transcriptionally upregulated during KRAS$^{G12}$-driven transdifferentiation of acinar cells, rendering the newly formed duct-like cells susceptible to PCD upon TAK1 inhibition. This

transcriptional upregulation of cell death regulators appears to be associated with the switch of acinar to ductal-like cell identity during ADM, and rather unaffected by KRAS$^{G12D}$ inhibition through MRTX1133 application.

## Concomitant inhibition of necroptosis and apoptosis restores ADM and PanIN formation in KRAS$^{G12D}$ TAK1$^{ΔAc}$ mice

Based on our in vitro data described above, we hypothesized that PCD induction in *Tak1*-deficient, KRAS$^{G12D}$-expressing transdifferentiated cells would eliminate premalignant lesions thereby inhibiting PDAC development in vivo. Contrary to our hypothesis, a previous study has demonstrated that the presence of necroptosis mediator RIPK3 promoted KRAS-driven PDAC progression in mice by eliciting an immunosuppressive response[27]. To assess the role of apoptosis and necroptosis induction in ADM/PanIN formation upon *Tak1* ablation in vivo, we generated KRAS$^{G12D}$ TAK1$^{ΔAc}$ mice with additional deletion of *Casp8* (KRAS$^{G12D}$ TAK1/CASP8$^{ΔAc}$), *Ripk3* (KRAS$^{G12D}$ TAK1$^{ΔAc}$ RIPK3$^{-/-}$) or both (KRAS$^{G12D}$ TAK1/CASP8$^{ΔAc}$ RIPK3$^{-/-}$). Immunoblotting analysis of transdifferentiated acinar cells upon TNF stimulation showed an impairment of MLKL phosphorylation in KRAS$^{G12D}$ TAK1$^{ΔAc}$ RIPK3$^{-/-}$ mice and KRAS$^{G12D}$ TAK1/CASP8$^{ΔAc}$ RIPK3$^{-/-}$ mice, verifying the inhibition of necroptosis, while cells from KRAS$^{G12D}$ TAK1/CASP8$^{ΔAc}$ mice and KRAS$^{G12D}$ TAK1/CASP8$^{ΔAc}$ RIPK3$^{-/-}$ mice showed an impairment of CASP3 cleavage, confirming the inhibition of apoptosis (Fig. 3a, b). Analyses of the three different mouse lines at 18 weeks of age showed that the additional deletion of *Casp8* or *Ripk3* alone had almost no effect on the occurrence of ADM. However, the combined deletion of *Casp8* and *Ripk3* re-established the formation of ADM/PanIN in KRAS$^{G12D}$ TAK1/CASP8$^{ΔAc}$ RIPK3$^{-/-}$ mice (Fig. 3c, d). This finding was confirmed in our 3D collagen matrix culture system, where only acinar cell explants from KRAS$^{G12D}$ TAK1/CASP8$^{ΔAc}$ RIPK3$^{-/-}$ mice were able to form stable ADM structures (Fig. 3e, f).

KRAS$^{G12D}$ TAK1/CASP8$^{ΔAc}$ RIPK3$^{-/-}$ mice exhibited ADM and PanIN lesions also at 52 weeks of age (Fig. 3g, h). However, unlike KRAS$^{G12D}$ mice, we did not detect any high-grade PanIN-3 stages or PDAC in KRAS$^{G12D}$ TAK1/CASP8$^{ΔAc}$ RIPK3$^{-/-}$ mice (Fig. 3g, h). This observation suggests that TAK1 controls ADM/PanIN formation through the regulation of PCD, but promotes late stages of PDAC oncogenesis through PCD-independent mechanisms.

Given that mutations are a driving force of PDAC development[28] and TAK1 might modulate chromosomal stability and the mutational spectrum of PDAC via NF-κB[29], we performed an array comparative genomic hybridization (aCGH) analysis on pancreatic tissue of five KRAS$^{G12D}$ TAK1/CASP8$^{ΔAc}$ RIPK3$^{-/-}$ mice and five KRAS$^{G12D}$ mice (Fig. 3i). Interestingly, both genotypes showed distinct chromosomal abnormalities with gain or loss of genetic regions. Further analysis of these chromosomal regions showed they indeed contained genes known to be involved in the pathogenesis of human PDAC[30], such as *TrpS3*, *Cdkn2a*, *Smad4*, *Rnf43* and *Arid1a* (Fig. 3j). However, we did not observe a consistent pattern of chromosomal/gene alterations in the

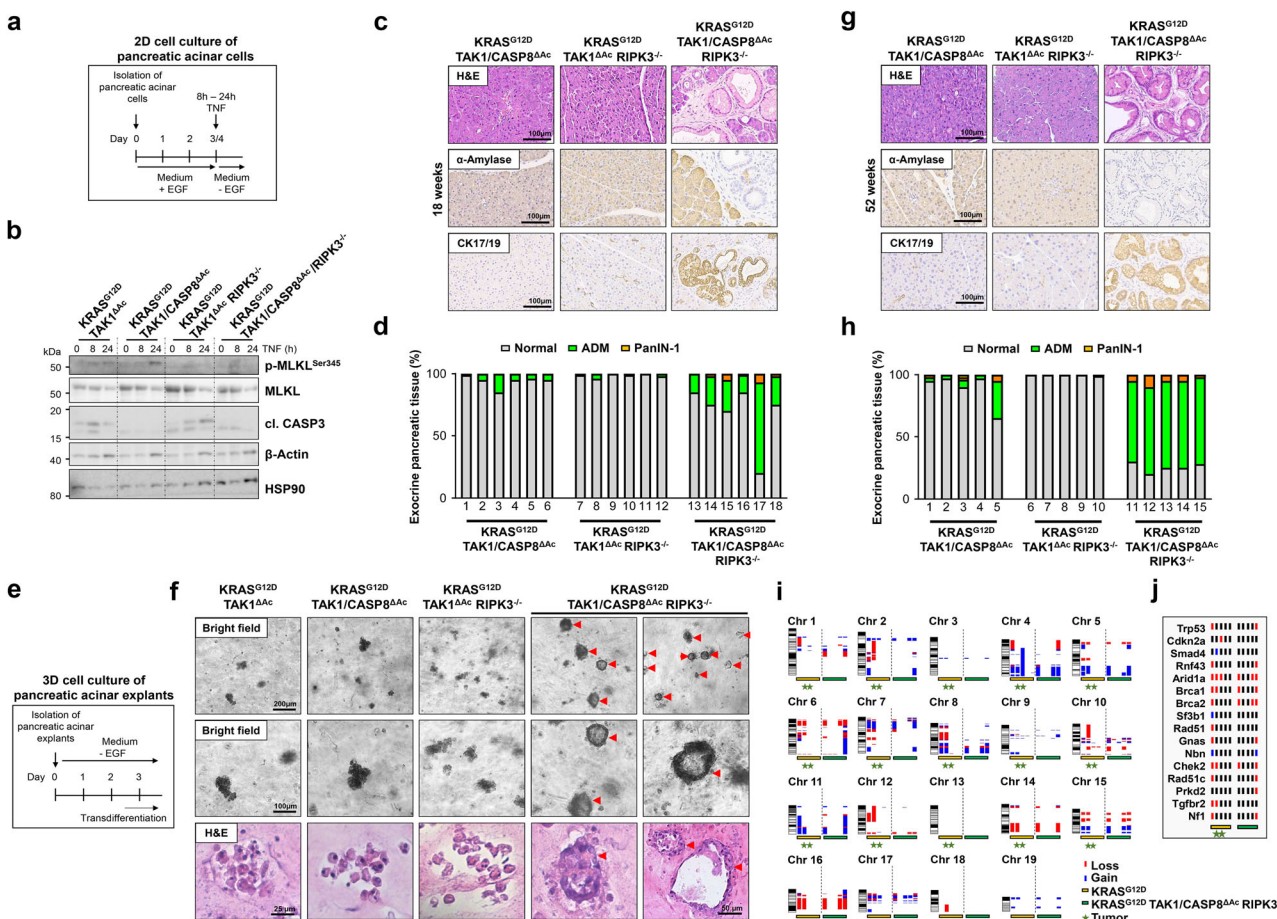

**Fig. 3 | Inhibition of necroptotic and apoptotic cell death restores transdifferentiation capability of pancreatic acinar cells from KRAS^G12D TAK1^ΔAc mice.**
**a, b** Experimental design to study cell death induction in primary pancreatic acinar cells grown in 2D cell culture upon treatment with TNF (100 ng/ml). Immunoblotting analysis in pancreatic acinar cells isolated from the indicated mice. The experiment was done once with one mouse per genotype. **c, d** H&E staining and IHC of α-Amylase and CK17/19 on pancreatic tissue sections from 18-week-old mice with the indicated genotype and quantification of healthy pancreas tissue (Normal), ADM and PanIN-1 of the same animals (*n* = 6 mice per genotype). **e, f** Experimental design to study the transdifferentiation capability of pancreatic acinar cell explants in 3D collagen matrices isolated from the indicated mice. Representative images are shown by bright-field microscopy and after H&E staining. Intact duct-like structures are highlighted with red arrowheads. The experiment was done with acinar explants from 2 different mice per genotype. **g, h** H&E staining and IHC of α-Amylase and CK17/19 of pancreatic tissue sections from 52-week-old mice with the

indicated genotype and quantification of healthy pancreas tissue (Normal), ADM and PanIN-1 of the same mice (*n* = 5 mice per genotype). **i, j** Array comparative genomic hybridization (aCGH) analysis of pancreas tissue (ADM, PanIN and PDAC areas) from 52-week-old WT (*n* = 3), KRAS^G12D (*n* = 5) and KRAS^G12D TAK1/CASP8^ΔAc RIPK3^-/- (*n* = 5) mice. The pancreatic tissue of individual transgenic mice was hybridized against the pancreatic tissue of age-matched WT mice and analyzed by aCGH. The q-arm of each chromosome is shown and chromosome numbers are indicated. Dark horizontal bars within the symbolized chromosomes represent G bands. Chromosomal deletions (loss) are indicated in red and amplifications (gain) in blue. Five mice per analyzed genotype are labeled by horizontal-colored bars. Mice with PDAC are labeled with a green star. Map of pancreatic cancer associated genes located in the chromosomal gain and loss regions detected by aCGH analysis is depicted in (**j**). Vertical lines next to each gene *locus* represent individual mice. Source data are provided in the Source Data file.

majority of samples/tumors between KRAS^G12D single transgenic mice and KRAS^G12D TAK1/CASP8^ΔAc RIPK3^-/- mice (Fig. 3i, j), arguing against the hypothesis that TAK1 deficiency mediated late stages of tumor promotion through changes in chromosomal stability.

Collectively, these data suggest that TAK1 inhibition prevents KRAS-driven PDAC development through the simultaneous induction of apoptosis and necroptosis during the early steps of ADM and PanIN formation in vivo. Moreover, the fact that only the combined inhibition of apoptosis and necroptosis re-established ADM formation following *Tak1* ablation underlines the plasticity existing in pancreatic duct cells regarding the choice of PCD pathway activation.

### TAK1 inhibition sensitizes PDAC patient-derived organoids to PCD
Our findings showed that TAK1 defines a decision point between cell survival and PCD during acinar-to-ductal transdifferentiation,

suggesting that TAK1 could be a promising pharmacological target in a chemopreventive setting[31]. However, given that PDAC is often diagnosed at an advanced stage, the selective action on KRAS^G12D-expressing transdifferentiated ductal-like but not on acinar cells suggests that TAK1 inhibition could also be an effective anti-tumor strategy at late stages of PDAC development. To test this hypothesis, we first treated three human PDAC cell lines with different KRAS mutational status (BxPC3^KRAS-WT, HPAC^KRAS-G12D and MIA-PaCa-2^KRAS-G12C) with 10 μM 5Z-7-Oxozeaenol, a concentration that was previously effective in killing murine transdifferentiated ductal cells leading to disintegration of the 3D duct-like structures (see Fig. 2d, e; Supplementary Fig. 2g). The efficiency of TAK1 inhibition was confirmed by qRT-PCR assessment of the expression of the NF-κB target genes *A20* and *IκBα* upon TNF stimulation, which was fully impaired in all cell lines (Supplementary Fig. 5a–c). Of note, treatment of the cells with 10 μM 5Z-7-Oxozeaenol for 48 h only led to a marginal cell death induction in all three PDAC

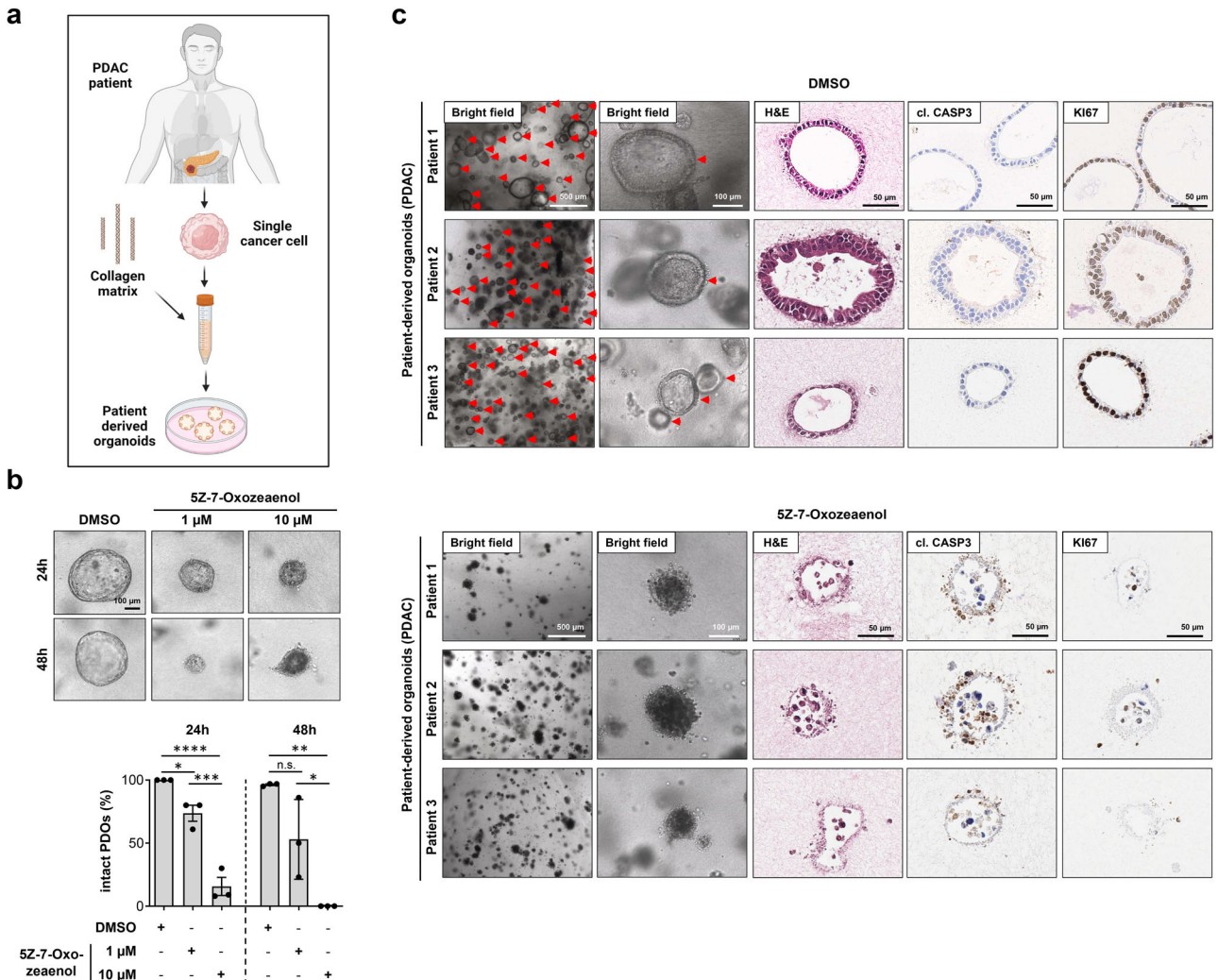

**Fig. 4 | Pharmacological TAK1 inhibition induces PCD in PDAC patient-derived tumor organoids. a** Schematic view of the generation of PDAC patient-derived tumor organoids (PDOs) from patient biopsies[70]. **b** Representative examples and quantification of intact PDOs after being established in collagen matrix cultures for 7 days and treated with DMSO (solvent) or 5Z-7-Oxozeaenol (1 μM or 10 μM) for additional 24 h and 48 h is indicated (% over the total PDOs examined). A total of $n = 21$, $n = 23$ and $n = 33$ PDOs (24 h) and $n = 108$, $n = 90$ and $n = 97$ PDOs (48 h) treated with DSMO, 1 μM 5Z-7-Oxozeaenol and 10 μM 5Z-7-Oxozeaenol were analyzed, respectively. Results are expressed as mean ± SEM. The experiment was done with PDOs established from the same three patients for all treatments ($n = 3$).

$P$ value was calculated by ordinary one-way ANOVA (two-tailed) with Tukey's multiple-comparisons test with n.s. = not significant, *$p = 0.0346$ (24 h), ****$p < 0.0001$ (24 h), ***$p = 0.0008$ (24 h), **$p = 0.0016$ (48 h), *$p = 0.0278$ (48 h). **c** Representative images of PDOs established from three different patients as in b, treated with DMSO (solvent) or 5Z-7-Oxozeaenol (10 μM) for 6 days with media changes every 48 h, and visualized by bright-field microscopy and after H&E staining and Ki67 and cl. CASP3 IHC, are shown ($n = 3$). Intact duct-like hollow structures are highlighted with red arrowheads. Source data are provided in the Source Data file.

cell lines (Supplementary Fig. 5d–f). However, combined treatment of the cells with 20 ng/ml TNF and 2 μM 5Z-7-Oxozeaenol induced significantly stronger cell death no matter whether the cells were bearing WT or mutant KRAS (Supplementary Fig. 5d–f). These results suggested that human PDAC cell lines, independent of their KRAS status, are rather resistant to TAK1 inhibition alone even at a relatively high concentration likely due to the increased mutational burden accumulated over the years.

We next performed pharmacological TAK1 inhibition experiments on patient-derived organoids (PDOs), a model system that more closely mimics the biological characteristics of the primary tumors[32,33]. To this end, we used PDOs isolated from three distinct PDAC patient donors (Supplementary Table 2)[34] and cultured them on floating 3D collagen matrices (Fig. 4a). Strikingly, incubation of all three PDO cultures with 5Z-7-Oxozeaenol led to their disintegration in a concentration- and time-dependent manner, indicating that similar to HPDEs, PDOs were more sensitive to TAK1 pharmacological inhibition

as a single treatment compared to the three PDAC cell lines (Fig. 4b). Additionally, we confirmed that TAK1 inhibition in these PDOs induced apoptosis, as assessed by H&E staining and cleaved CASP3 immunostaining, and this was associated with a marked reduction in the cell proliferation marker Ki-67 (Fig. 4c). Together, these data show that TAK1 inhibitors could also be effective inducers of cell death in organoids established from freshly-isolated samples of patients with advanced PDAC.

## Inhibition of TAK1 does not result in a pro-inflammatory immune response

Specific immunological responses can promote or inhibit tumorigenesis depending on the experimental context. RIPK3 was shown to inhibit T cell infiltration by inducing an immune-suppressive microenvironment in KRAS-driven PDAC mouse model[27], while our in vivo data rather suggested that activation of the necrosome inhibited PDAC development. To investigate whether a possible immunoregulatory

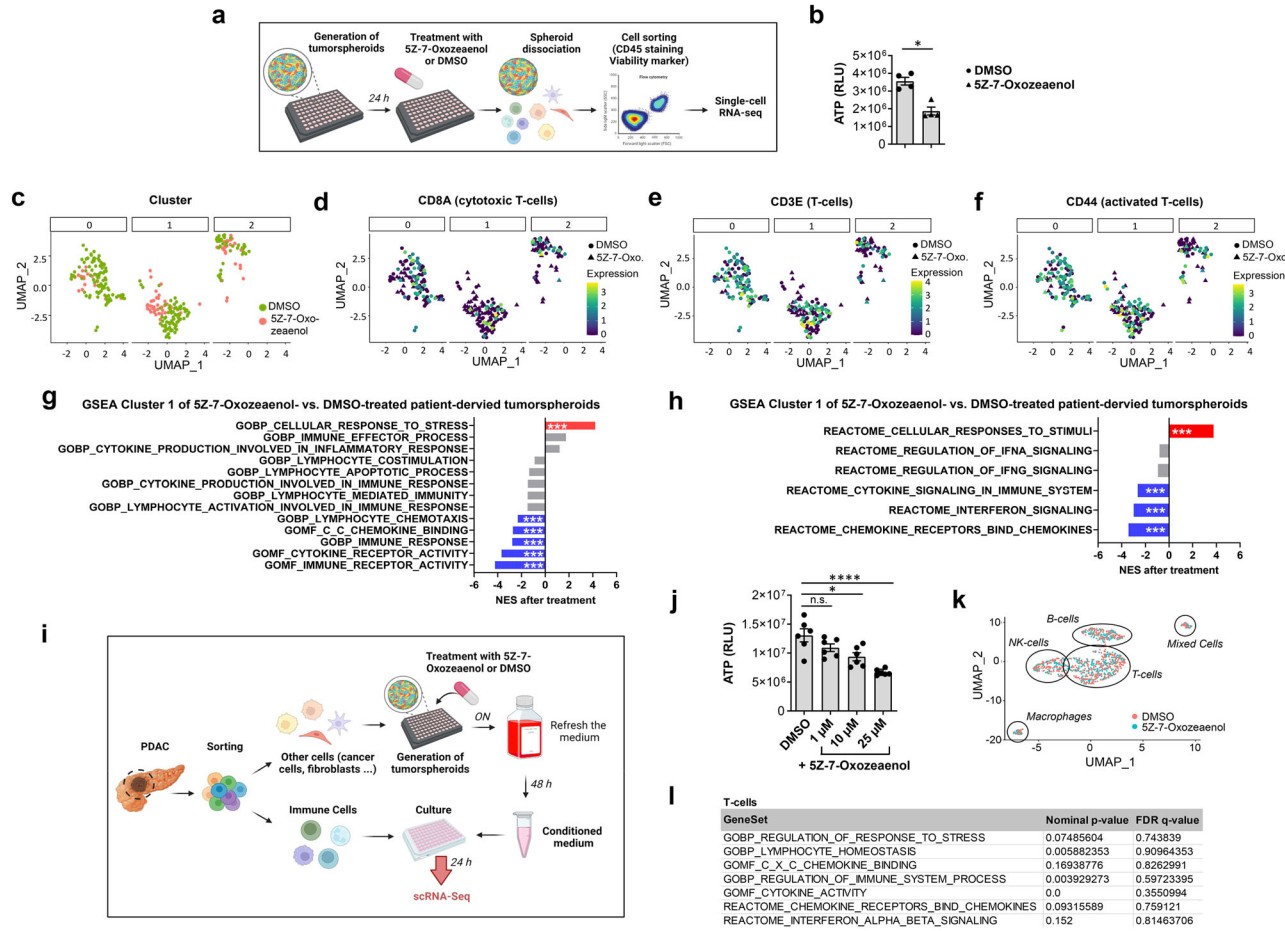

**Fig. 5 | TAK1 inhibition does not cause a pro-inflammatory immune response.**
**a** Experimental approach to assess the immune response elicited by ubiquitous TAK1 inhibition on tumor spheroids generated by isolation of total cell populations from tumor tissue of one PDAC-patient[71]. **b** Cell viability assay of PDAC-derived tumor spheroids (*n* = 4 per condition) treated with DMSO or 5Z-7-Oxozeaenol (25 μM) for 72 h. Results are expressed as mean ± SEM. *P* value was calculated by Mann–Whitney U test (two-tailed), *p* = 0.0286. **c** UMAP of single-cell transcriptomes showing DMSO and 5Z-7-Oxozeaenol-treated immune cells (DMSO: *n* = 208; 5Z-7-Oxozeaenol: *n* = 78). **d–f** UMAP of CD8+ (DMSO: *n* = 67; 5Z-7-Oxozeaenol: *n* = 7), CD3E+ (DMSO: *n* = 117; 5Z-7-Oxozeaenol: *n* = 22) and CD44+ (DMSO: *n* = 122; 5Z-7-Oxozeaenol: *n* = 23) T-cells, isolated from PDAC-derived tumor spheroids, indicating T-cells as the main cell population. Color bar indicates log2-normalized expression. **g**, **h** Gene Set Enrichment Analysis (GSEA) in cluster 1 of T-cells isolated from DMSO vs. 5Z-7-Oxozeaenol-treated PDAC patient-derived tumor spheroids. Normalized enrichment score (NES) of significantly enriched (red), suppressed (blue) or non-significantly regulated (gray) (FDR > 0.05)

pathways after 5Z-7-Oxozeaenol treatment (***FDR q < 0.001) are presented.
**i** Experimental approach to assess the immune response elicited upon TAK1 inhibition on CD45+ cell-depleted tumor spheroids generated by isolation and in vitro reconstitution from tumor tissue of one PDAC-patient[72]. **j** Cell viability assay of PDAC-derived tumor spheroids (*n* = 6 per condition) treated with DMSO or 5Z-7-Oxozeaenol (1 μM, 10 μM and 25 μM) for 72 h. Results are expressed as mean ± SEM. *P* value was calculated by ordinary one-way ANOVA (two-tailed) with Tukey's multiple-comparisons test with n.s. = not significant, *p* = 0.0114, ****p* < 0.0001. **k** UMAP of single-cell transcriptomes of PDAC-derived immune cells subjected with the supernatant of DMSO- or 5Z-7-Oxozeaenol (25 μM)-treated tumor spheroids (DMSO: *n* = 529; 5Z-7-Oxozeaenol: *n* = 427). **l** GSEA of PDAC-derived T-cells incubated with the conditioned medium from DMSO- or 5Z-7-Oxozeaenol-treated tumor spheroids. NES was non-significantly altered (FDR > 0.05) in all pathways that were affected after 5Z-7-Oxozeaenol treatment in g-h. Source data are provided in the Source Data file.

effect of TAK1 inhibition could explain this discrepancy, we generated tumor spheroids (containing a mix of tumor and immune cells) derived from another PDAC patient and treated them with DMSO or 5Z-7-Oxozeaenol for 24 h (Fig. 5a, Supplementary Table 3). Similar to our results in PDOs, TAK1 inhibition significantly impaired spheroid cell viability (Fig. 5b). To assess the activation state of the immune cells, CD45+ leukocytes were isolated from the DMSO- or 5Z-7-Oxozeaenol-treated tumor spheroids and analyzed by scRNA-Seq applying the SORT-seq technology[35] (Fig. 5a, Supplementary Fig. 6). Cell clustering was based on similar transcriptome profiles and revealed three clusters of activated CD3+, CD8+ and CD44+ cytotoxic T cells (Fig. 5c–f; Supplementary Table 4). In contrast, CD4+ T helper cells, CD14+ monocytes, CD68+ macrophages, and FOXP3+ Tregs could not be detected (Supplementary Fig. 7a; Supplementary Table 4). Out of the three identified T cell clusters, only cluster 1 showed significant

transcriptomic changes as a result of TAK1 inhibitor treatment (Supplementary Data 1). Gene set enrichment analysis (GSEA) of the differentially expressed genes revealed that 5Z-7-Oxozeaenol significantly suppressed key pathways associated with cellular immune responses, such as cytokine and interferon signaling and immune receptor activity, while it led to increased expression of markers involved in cellular response to stress (Fig. 5g, h; Supplementary Fig. 7b; Supplementary Data 1). These data suggest that TAK1 inhibition did not induce strong adaptive immune responses, despite inducing the death of tumor cells in PDAC-derived spheroids. Considering that TAK1 is essential for NF-κB activation, the immunologically silent death of TAK1-inhibited spheroids could reflect the impaired production and release of pro-inflammatory cytokines from dying cancer cells, but it could also be the result of an impairment of T cell activation that is required for their proinflammatory response.

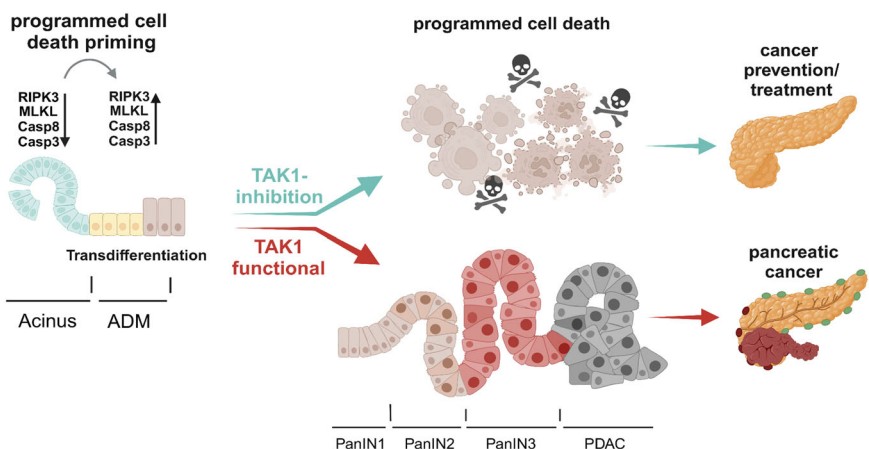

**Fig. 6 | Proposed model on the role of TAK1 inhibition in KRAS-driven ADM and PDAC development.** KRAS-dependent, and likely KRAS-independent, ADM induction leads to upregulated expression of PCD mediating molecules. Through its NF-κB-independent prosurvival functions, TAK1 prevents elimination of the PCD-primed transdifferentiated ductal cells, thereby enabling PanIN formation and PDAC development. In contrast, TAK1 deficiency/inhibition impairs cell survival during ADM and prevents PanIN establishment and progression to PDAC[73].

To avoid the concomitant pharmacological inhibition of TAK1 in immune cells, an additional experiment was carried out in which the CD45⁺ immune cells were first separated from the remaining cells (e.g. cancer cells, fibroblasts, etc.) (Fig. 5i, Supplementary Fig. 8, Supplementary Table 5). The immune cell-depleted spheroids were then treated overnight with 5Z-7-Oxozeaenol or DMSO, followed by removal of the inhibitor and culture in 5Z-7-Oxozeaenol-free medium for 48 h to obtain a conditioned medium containing factors released by the dying cancer cells. The conditioned medium was used to stimulate the sorted immune cells, which were subsequently processed for scRNA-Seq analysis (Fig. 5i, Supplementary Fig. 8). In line with the previous experimental setting, TAK1 inhibition strongly impaired the spheroid viability in a concentration-dependent manner (Fig. 5j). Based on their expression profiles, four distinct immune cell clusters could be detected (T cells, B cells, NK cells and macrophages) and the clustering was not affected by the treatment (Fig. 5k; Supplementary Fig. 9a, b). Interestingly, differential gene expression analysis showed no difference between the two treatment groups (Fig. 5l, Supplementary Fig. 9c, Supplementary Data 2). Accordingly, the pathways involved in inflammation and response to cellular stress that were differentially regulated in the GSEA analysis of the first experimental setup (Fig. 5g, h) were not significantly altered anymore (Fig. 5l, Supplementary Data 2), confirming that TAK1 inhibition in PDAC-derived spheroids did not elicit significant inflammatory responses.

## Discussion

PDAC is one of the most deadly cancers with lethality close to incidence. Therefore, understanding the mechanisms underlying pancreatic carcinogenesis is essential for the development of novel therapies[2]. Our present study (summarized in Fig. 6) revealed that upregulation of PCD regulators, namely the apoptotic caspases CASP8 and CASP3 and the necroptosis mediators RIPK3 and MLKL, are part of the transcriptional rewiring induced during acinar-to-ductal transdifferentiation process, a key step in KRAS-dependent carcinogenesis[4]. This upregulation of PCD mediators sensitizes transdifferentiated cells to both apoptosis and necroptosis, a feature that is counterbalanced by TAK1 kinase, which represents a promising target to induce the elimination of premalignant as well as cancer cells.

TAK1 is activated by several inflammatory and stress-related pathways and has cytoprotective and pro-inflammatory functions. For instance, TAK1 activates canonical NF-κB signaling through phosphorylation and activation of IKKβ[10]. In addition, TAK1 suppresses RIPK1 kinase activity-dependent cell death via distinct inhibitory

phosphorylation steps at multiple RIPK1 serine residues, which are catalyzed by the TAK1 targets mitogen-activated kinase 2 (MK-2)[36] and IKKα/IKKβ[37,38]. Our in vivo data showing that NF-κB inhibition through *Rela* ablation did not phenocopy the anti-tumorigenic effect of *Tak1* ablation, and even slightly enhanced PDAC development, demonstrated that TAK1 promotes pancreatic tumorigenesis in an NF-κB-independent way. On the contrary, the use of TAK1 and RIPK1 kinase inhibitors in our in vitro experiments suggested that the lack of TAK1-mediated negative regulation on RIPK1 kinase activity was primarily responsible for PCD induction during ADM and likely prevented cancer development in KRAS^{G12D} TAK1^{ΔAC} mice. This mechanism was further supported by the reversal of the anti-PDAC effect by the combined inhibition of CASP8-mediated apoptosis and RIPK3-dependent necroptosis in these mice. The TAK1 paradigm could also provide a possible explanation for the previously described opposite effects of deleting the IKK subunit *Ikkb*[39] versus the NF-κB subunit *Rela*[21] in KRAS-driven PDAC development.

Activation of RAS signaling is found in up to 90% of human PDAC[3]. It could be therefore clinically relevant to underline that the effects we observed in mice, 3D mouse pancreatic organoid cultures, and human PDAC organoids and spheroids upon genetic or pharmacological TAK1 inhibition were all obtained in the context of KRAS-driven transdifferentiation and PDAC development. Interestingly, the upregulated expression of cell death genes during the in vitro transdifferentiation of acinar to ductal cells was not affected by MRTX1133, a selective KRAS^{G12D} inhibitor that has recently shown promising anti-tumor efficacy in PDAC preclinical mouse models[24] and has entered clinical trials in patients with advanced solid tumors (NCT05737706). Despite not regulating the expression of cell death mediators, MRTX1133 was effective in inhibiting duct-like structure formation. Constitutive activation of KRAS is known to lead to irreversible pancreatic ADM, but this process is also regulated by multiple other signaling pathways, including Notch, Hedgehog, Wnt, EGFR, NF-κB, and TGFβ[4,6]. Although MRTX1133 does inhibit MEK/ERK and PI3K/AKT/mTOR activation downstream of KRAS^{G12D} [23], it remains unclear to what extent it affects other pathways involved in ADM. Our results suggest that the elevated expression of PCD-associated molecules in the in vitro ADM assay is independent of MRTX1133 and putatively RAS activation, although an incomplete KRAS^{G12D} inactivation by the inhibitor could not be formally excluded. This is consistent with our data showing that the KRAS mutational status in immortalized HPDEs and PDAC cancer cell lines was not associated with consistent differences regarding their sensitivity to cell death upon TAK1 inhibition with or without TNF. Instead,

we propose that this PCD priming is a common event during the dedifferentiation of acinar cells and their transdifferentiation to duct-like progenitor cells independent of the underlying signaling event(s) driving this process. The role of TAK1 in transdifferentiated ductal cells is to act as a rheostat: when it is functional, it promotes survival of the primed cells and their progression to PanINs and PDAC, while when it is inactivated, it promotes RIPK1-dependent PCD, most probably triggered by molecules that are present during ADM, such as TNF or TGFβ (Fig. 6).

The transcription factor Sp1 and the epigenetic regulator UHRF1, which have both been involved in the transcriptional control of RIPK3[40], are overexpressed in PDAC patient samples and pancreatic cancer cell lines[41,42], suggesting that they could also be implicated in the regulation of cell death mediators during transdifferentiation. Additionally, the presence of multiple NF-κB and STAT binding sites in the promoter sequences of the upregulated genes suggests that NF-κB and/or IL-6/STAT3 pathways could be involved. Interestingly, RIPK3 upregulation appears to be a general phenomenon associated with ductal differentiation in the hepatobiliary and intestinal tract, as we and others have shown that cholangiocytes express much higher RIPK3 levels than hepatocytes, thereby being susceptible to necroptosis induction[12,14]. This process may reflect the fact that pancreatic and hepatic ductal cells, similar to intestinal epithelial cells that also express high levels of RIPK3[43], are more likely to come into contact with microbes or microbial components, such as LPS, than parenchymal pancreatic acinar cells or hepatocytes.

Cell death induction in the course of inflammatory diseases and carcinogenesis can have opposing outcomes. Typically, tissue injury at an early stage can lead to persistent inflammation promoting cancer development, whereas in established tumors it mediates cancer cell elimination and anti-tumor immune responses. In addition, it is increasingly clear that not only PCD levels and modalities are important for the outcome, but also what is released by the dying cells, namely cytokines or damage-associated molecular patterns (DAMPs)[44]. Along this line, genetic induction of ferroptosis was shown to aggravate KRAS-driven PDAC in mice[45], while it inhibited PDAC growth when induced after cancer establishment[46]. Our mouse genetic data showed that apoptosis and necroptosis appear to be additional PCD pathways influencing pancreatic carcinogenesis, although their induction at an early stage protected against PDAC development. Intriguingly, a study addressing the role of RIPK3 in KRAS$^{G12D}$-mediated PDAC development showed that RIPK1/RIPK3 signaling promoted tumorigenesis despite inducing necroptosis. This pro-carcinogenic role of RIPK3 was attributed to a necroptosis-independent function that elicited macrophage-induced anti-tumor immune suppression through CXCL1/Mincled signaling[27]. On the basis of these data suggesting a pro-carcinogenic function of RIPK1/RIPK3 signaling, a clinical trial testing RIPK1 inhibitors in PDAC patients (ClinicalTrials.gov identifier: NCT03681951) was initiated but was discontinued at an early phase. A possible explanation for the opposing effect of PCD induction between our in vivo data and the aforementioned study[27] could be the presence or absence of TAK1 kinase in transdifferentiated cells. As mentioned above, TAK1 is essential for NF-κB-mediated synthesis and release of cytokines and chemokines from living and dying cells. Indeed, a key determinant for the ability of apoptotic or necroptotic cancer cells for efficient cross-priming of CD8$^+$ T cells was shown not to be their type of death but rather their competence for NF-κB−driven gene expression[47]. Accordingly, our group has recently shown that NF-κB inhibition promotes lethal necroptosis in the liver preventing cytokine release and protecting against hepatocellular carcinoma[48]. Similarly, our in vivo data presented here suggest that TAK1-deficient KRAS$^{G12D}$-expressing transdifferentiated ductal cells likely die without releasing high levels of pro-inflammatory molecules, thereby undergoing an immunologically silent death. This was also reflected in our human spheroid experiments when tumor and immune cells were

either treated simultaneously with the TAK1 inhibitor or when only the tumor cells were treated and the conditioned medium was used to activate the immune cells that were sorted out from the patient PDAC sample. In both experimental setups, scRNA-Seq analysis showed that pharmacological inhibition of TAK1 did not elicit significant immune responses (Fig. 5). A contribution of the NF-κB activating function of TAK1 could also explain the fact that KRAS$^{G12D}$ TAK1/CASP8$^{ΔΔC}$ RIPK3$^{-/-}$ mice developed PanIN lesions, which did not progress to advanced PDAC. Previous studies have also associated NF-κB activation at late stages of PDAC progression with regulating senescence or inflammation[21,39].

In summary, our study suggests that inducing rather than preventing PCD through TAK1 inhibition could hold a great translational potential as an anti-cancer strategy for the prevention and treatment of PDAC. TAK1 inhibition could be applied to kill cells during the initial acinar-to-ductal transdifferentiation process and prevent PanIN development before PDAC establishment. Considering the lack of effective screening and prevention approaches in PDAC[49], this approach could be particularly appealing as an effective chemoprevention strategy in high-risk patients (e.g. chronic pancreatitis, genetic high-risk patients, etc[49,50]) and thereby reduce the incidence and subsequent mortality of PDAC. In addition, TAK1 inhibition effectively induced tumor cell death in already established PDAC cells. Importantly, the sensitization of pancreatic cells to apoptosis and necroptosis upon TAK1 inhibition was specific for KRAS-driven transdifferentiated cells, since KRAS$^{WT}$ TAK1$^{ΔΔC}$ mice developed no gross pancreatic phenotype. This predicts that TAK1 inhibitors in a PDAC setting would preferentially target cells that have undergone ADM and oncogenic transformation, while they would spare the untransformed pancreatic cells. However, as systemic use of TAK1 inhibitors is likely to disturb homeostasis in other organs, the development of targeted delivery methods, for example using "smart nanoparticles[51,52], could minimize the adverse effects. Finally, the fact that combined deletion/inhibition of TAK1 and Caspase-8 still blocked transdifferentiation and PDAC development though activation of necroptosis suggests that TAK1 inhibition could still be an effective anti-tumor therapy in PDACs that show a downregulation of the apoptosis machinery in addition to KRAS activation[53]. Altogether, our results demonstrate the potent tumor-suppressing effect of TAK1 inhibition and provide a rationale for conducting preclinical and clinical trials in the future.

## Methods
### Generation of genetically modified mouse models
LSL-KRAS$^{G12D/+}$, Ptf1a-cre, Tak1$^{fl/fl}$, Ripk3$^{-/-}$, Casp8$^{fl/fl}$ and RelA$^{fl/fl}$ strains were interbreed to obtain LSL-KRAS$^{G12D/+}$ Ptf1a-cre (termed KRAS$^{G12D}$), LSL-KRAS$^{G12D/+}$ Tak1$^{fl/fl}$ Ptf1a-Cre (termed KRAS$^{G12D}$ TAK1$^{ΔΔc}$), Tak1$^{fl/fl}$ Ptf1a-Cre (termed TAK1$^{ΔΔc}$), LSL-KRAS$^{G12D/+}$ Tak1$^{fl/fl}$ Ripk3$^{-/-}$ Ptf1a-Cre (termed KRAS$^{G12D}$ TAK1$^{ΔΔc}$ RIPK3$^{-/-}$), LSL-KRAS$^{G12D/+}$ Tak1$^{fl/fl}$ Casp8$^{fl/fl}$ Ptf1a-Cre (termed KRAS$^{G12D}$ TAK1/CASP8$^{ΔΔc}$), LSL-KRAS$^{G12D/+}$ Tak1$^{fl/fl}$ RIPK3$^{-/-}$ Casp8$^{fl/fl}$ Ptf1a-Cre (termed KRAS$^{G12D}$ TAK1/CASP8$^{ΔΔc}$ RIPK3$^{-/-}$), LSL-KRAS$^{G12D/+}$ RelA$^{fl/fl}$ Ptf1a-Cre (termed KRAS$^{G12D}$ RelA$^{ΔΔc}$)[17,18,21,54–56]. Mice were bred on a mixed C57/BL6 - SV129Ola genetic background. In all experiments, littermates carrying the respective loxP-flanked alleles but lacking expression of Cre recombinase were used as wild-type (WT) controls. Age-, gender-, and equal average tumor volume-matched mice were randomly assigned to groups, based on their genotypes, and experiments were not blinded. Both male and female mice are included in all groups. A precalculation of the in vivo mouse sample sizes was performed and approved to ensure an optimal balance between the animal welfare guidelines and a reasonable sample number for the experiments. Sample sizes and mice age is indicated in the figure legends. All animal experiments were approved by the Federal Ministry for Nature, Environment and Consumers' Protection of the state of North Rhine-Westphalia and were performed in accordance to the respective national, federal, and institutional regulations.

The maximal permitted tumor size of ≥ 1.5 cm was not exceeded. Mice were housed in individually ventilated cages (IVC) with HEPA-filter from Tecniplast at $22 \pm 2\,°C$, with a humidity of $55 \pm 10\%$, and an air exchange rate of 75 times on a continuous 12 h light-dark cycle from 6 am to 6 pm.

### Human pancreatic tissue microarray (TMA)

Human pancreatic tissue samples were provided by the tissue bank of the National Center for Tumor Diseases Heidelberg (NCT, Heidelberg, Germany) in agreement with the regulations of the tissue bank and local Ethics Committee of the University of Heidelberg approval (no. 206/2005). The project conducted in accordance with the ethical standards laid down in the Declaration of Helsinki. Patients have given their informed consent without being paid. The gender of the participants was determined on the basis of self-reports. No gender was excluded. All patients received standard surgical resection. Paraffin embedded tissues were preprocessed by a pathologist after surgical resection and confirmed as PDAC prior to further investigation.

A tissue microarray (TMA) was generated with representative tumor areas (duplicates, core diameter 1 mm). The basic characteristics of the PDAC patient samples included in the TMA are shown in Supplementary Table 1.

### Isolation of murine primary pancreatic acinar cells, three-dimensional (3D) collagen matrix culture and two-dimensional (2D) cell culture, in vitro transdifferentiation, stimulation and quantification

The isolation of primary pancreatic acinar cells was performed using a rapid isolation protocol[22] with some modifications. Five- to seven-week-old mice of both genders were sacrificed, the pancreas was harvested, chopped into small pieces and digested in a collagenase solution (1x HBSS (PAN-Biotech, cat no: P04-34500), 10 mM HEPES (Carl Roth; cat no: HN78.3), 400 U/mL Collagenase from *Clostridium histolyticum* (Sigma, cat no: C0130) and 0.25 mg/ml Trypsin inhibitor from soybean (Sigma, cat no: T6522)) for 20–30 min at $37\,°C$ under 5% (v/v) $CO_2$ atmosphere. During this time, a mechanical dissociation was performed every 5–7 min using serological pipettes of decreasing size (25, 10, and 5 ml). The digestion was stopped by adding cold buffered washing solution (1x HBSS (PAN-Biotech, cat no: P04-34500) containing 5% Fetal Bovine Serum (FBS, PAN-Biotech, cat no: P30-3033) and 10 mM HEPES (Carl Roth; cat no: HN78.3)) and cells were washed three times with 10 ml washing solution. The cell pellet was resuspended in culture medium (Waymouth's MB752/1 medium (Gibco, cat no: 11220035) supplemented with 2.5% FBS (PAN-Biotech, cat no: P30-3033), 1% Penicillin-Streptomycin solution (PAN-Biotech, cat no: P06-07100), 1x Insulin-Transferrin-Selenium solution (ITS) (PAN-Biotech, cat no: P07-03210), 0.25 mg/ml Trypsin inhibitor from soybean (Sigma, cat no: T6522) +/- 25 ng/ml Epidermal Growth Factor (EGF, Gibco, cat no: AF-100-15)) and cell suspension was passed through $100\,\mu m$ filter. The cell number was counted, and the cells were allowed to recover for about 3 h at $37\,°C$ under 5% (v/v) $CO_2$ atmosphere.

For 2D cell culture, the isolated primary acinar cells were transferred into type I collagen-coated (50 µg/µml, Merck Millipore, cat no: 08-115) in 0.02 M acetic acid (Carl Roth, cat no: 6755.1) 6-well culture dishes and cultured in the presence of EGF (Gibco, cat no: AF-100-15) at $37\,°C$ under 5% (v/v) $CO_2$ atmosphere. Once the acinar cells spread and lost their acinar morphology, the medium was replaced by medium without EGF (Gibco, cat no: AF-100-15), and the cells were stimulated with 100 ng/ml TNF (Peprotech, cat no: 315-01 A) for the indicated time points. For the time point labeled '0', cells were cultured 1 h with culture medium without EGF (Gibco, cat no: AF-100-15) and TNF (Peprotech, cat no: 315-01 A).

For 3D collagen matrix cultures, 48-well culture dishes were coated with rat tail type I collagen (2.5 mg/ml, Merck Millipore, cat no: 08-115) in Waymouth's MB752/1 medium (Gibco, cat no: 11220035). The cells that were resuspended in the culture medium without EGF (Gibco, cat no: AF-100-15) were then mixed with equal volumes of neutralized rat tail type I collagen (Merck Millipore, cat no: 08-115), plated on top of collagen-coated wells and incubated at $37\,°C$ under 5% (v/v) $CO_2$ atmosphere for about 30 min. After solidification, culture medium was added on top and refreshed on the next day. Inhibitors (5Z-7-Oxozeaenol (10 µM, Sigma, cat no: O9890-1MG), Z-VAD-*fmk* (25 µM, Merck Millipore, cat no: V116), Nec-1s (50 µM, Biovision, cat no. 2263-5) or TPCA-1 (10 µM, Tocris Bioscience, cat no: 2559)) or solvent (DMSO, Carl Roth, cat no. 7029.1) were added to the culture medium at the indicated time points. Z-VAD-*fmk* and Nec-1s were added 2 h before 5Z-7-Oxozeaenol. For bright field microscopy, cells and duct-like structures were imaged with a Leica DM IL LED microscope (Leica). The total amount of visible duct-like structures in 3D collagen matrices was counted for each condition per well.

### Primary KRAS^G12D-expressing pancreatic acinar cell isolation and 3D collagen matrix culture in vitro for RNA isolation and LDH assay

Three 4-6-week-old KRAS^G12D mice of both genders were sacrificed, their pancreata were harvested, chopped into small pieces and digested twice with 1.2 mg/mL Collagenase from *Clostridium histolyticum* (Sigma, cat no: C0130) dissolved in McCoy's 5 A medium (Gibco, cat no:16600082) containing 0.02% Trypsin inhibitor from soy bean (Sigma, cat no: T6522) and 0.1% BSA (Sigma, cat no. A9418) for 10 min each. Cells were passed through a $100\,\mu m$ mesh and washed with McCoy's 5 A medium medium (Gibco, cat no: 16600082) and spun down at 14x g. Afterwards, cells were recovered in culture medium (Waymouth's MB752/1 medium (Gibco, cat no: 11220035) containing 0.1% BSA (Sigma, cat no. A9418), 0.1% FCS (Gibco, cat no: A5670701), 0.01% Trypsin inhibitor from soy bean (Sigma, cat no: T6522), 1x Insulin-Transferrin-Selenium (PAN-Biotech, cat no: P07-03210), 50 µg/mL Bovine Pituitary Extract (Gibco, cat no: 13028014), 10 mM HEPES (Carl Roth; cat no: HN78.3), 2.6 mg/mL NaHCO3 (Merck, cat no: 106329)) and incubated for 30-60 min at $37\,°C$. At the endpoint of the incubation, defined as time point 'd0', samples were collected for RNA isolation.

Isolated acinar cells were recovered in culture medium, mixed at a ratio of 1:1 with neutralized collagen I from rat tail (Corning, cat no: CLS354236) and plated on a solid layer of 2.5 mg/mL collagen I from rat tail (Corning, cat no: CLS354236) containing 10% 10x PBS (PAN-Biotech, cat no: P04-53500). After solidification, an additional layer of 2.5 mg/mL collagen and 10x PBS (PAN-Biotech, cat no: P04-53500) was added. Culture medium ±MRTX1133 (200 nM or 400 nM, MedChemExpress, cat no: HY-134813) was added on top of the three-layer culture system. On the next day (d1), the medium was refreshed. To determine the duct-like structure rate at day 3 (d3), all grape-like structures were counted as acini, while hollow, spherical structures were identified as ducts. The duct-like structure rate was calculated as the ratio of duct-like structures to the total number of identified structures according to the formula: duct-like structure rate (%) = duct-like structures/total number of identified structures *100. Cells for RNA isolation were collected at d3.

Cell death was estimated using an LDH release-based cytotoxicity assay (Roche, cat no: 11644793001) according to the manufacturer's protocol. Supernatants were collected during the medium change on d1 and at the endpoint d3. Freshly isolated acinar cells (d0) lysed with 2% Triton X-100 (Sigma, cat no: X100) were used as the positive control (PC). Data were generated using i-control 2.0 (for infinite reader) software. The percentage of dead cells was calculated according to the following formula: Cell death = $(Abs_{sample} - Abs_{blank})/(Abs_{PC} - Abs_{blank})$.

### Cell lines and assessment of cell death by LDH assay

The human PDAC cell line BxPC-3 (KRAS^WT, Sanger Cell Lines Project, cat no: COSS906693) was cultured in RPMI Medium 1640 (Gibco, cat

no: 11875085) supplemented with 10% Fetal Bovine Serum Premium (PAN-Biotech, cat no: P30-1302), 1% Penicillin-Streptomycin solution (PAN-Biotech; cat no: P06-07100) and 2 mM L-Glutamine (PAN-Biotech, cat no: P04-80050). The human PDAC cell line HPAC (KRAS$^{G12D}$, Sanger Cell Lines Project, cat no: COSS1298136) was cultured in DMEM/F-12 with 15 mM Hepes (Carl Roth; cat no: HN78.3) and Sodium bicarbonate without L-glutamine (Sigma, cat no: D6421) supplemented with 5% Fetal Bovine Serum Premium (PAN-Biotech, cat no: P30-1302) and 1% Penicillin-Streptomycin (PAN-Biotech, cat no: P06-07100). The human PDAC cell line MIA-PACA-2 (KRAS$^{G12C}$, ATCC, cat no: CRL-1420) was cultured in DMEM (PAN-Biotech, cat no: P04-03590) supplemented with 2 mM L-Glutamine (PAN-Biotech, cat no: P04-80050), 10% Fetal Bovine Serum (PAN-Biotech, cat no: P30-3033) and 1% Penicillin-Streptomycin solution (PAN-Biotech, cat no: P06-07100). HPDE cells (generously provided by the laboratory of Dr. Anil K. Rustgi, New York, USA) were cultured in keratinocyte serum-free medium supplemented with bovine pituitary extract (Gibco, cat no: 13028014), EGF (Gibco, cat no: AF-100-15) and 1% Penicillin-Streptomycin solution (PAN-Biotech, cat no: P06-07100). For the expression of human KRAS$^{G12D}$ or GFP under a Doxycycline-inducible promoter, the cells were transduced with pINDUCER lentiviral system[57] followed by Hygromycin selection. To activate the promoter, HPDE cells were treated with doxycycline (400 ng/ml, Merck Millipore, cat no: D5207) for three days before further treatments were performed. For RNA isolation, cells were seeded in 12-well culture dishes, allowed to adhere overnight and treated with DMSO (Carl Roth, cat no: 7029.1) or 5Z-7-Oxozeaenol (10 μM, Sigma, cat no: O9890-1MG) for 1 h before human TNF (20 ng/ml, Merck Millipore, cat no: SRP3177) was added to the medium for another h. For the LDH-assay, cells were seeded in 48-well culture dishes, allowed to adhere overnight and treated with DMSO (Carl Roth, cat no: 7029.1), 5Z-7-Oxozeaenol (10 μM, Sigma, cat no: O9890-1MG), human TNF (20 ng/ml, Merck Millipore, cat no: SRP3177) or TNF (20 ng/ml, Merck Millipore, cat no: SRP3177)/5Z-7-Oxozeaenol (2 μM, Sigma, cat no: O9890-1MG) for 48 h.

Cell death was estimated using the LDH-based, CytoTox 96 cytotoxicity assay (Promega, cat no: G1781) according to the manufacturer's protocol. LDH ratio (released vs. total LDH) was measured on samples of cell supernatant before and after cell lysis with 1% Triton X-100 (Sigma, cat no: X100) using i-control 2.0 (for infinite reader) software. The measurements were performed in duplicates in 3 independent experiments.

### RNA isolation and quantitative real-time polymerase chain reaction (qRT-PCR)

Total RNA from 2D culture of primary pancreatic acinar explants or human pancreatic cancer cell lines was isolated at the indicated time points using the NucleoSpin® RNA Plus kit (Macherey-Nagel, cat no: 740984.50) according to the manufacturer's protocol. Total RNA from 3D collagen matrix culture of primary pancreatic acinar explants was isolated using Maxwell® 16 LEV simply RNA Tissue Kit (Promega, cat no: AS1280) and Maxwell® 16 Instrument (Promega, cat no: AS2000) following the manufacturer's instructions. Freshly isolated acinar cells (d0) were directly lysed for RNA isolation, collagen-embedded acinar cells (d3) were first released from the collagen gel using 1.2 mg/mL Collagenase from *Clostridium histolyticum* (Sigma, cat no: C0130) in McCoy's 5 A Medium medium (Gibco, cat no:16600082) containing 0.02% Trypsin inhibitor from soy bean (Sigma, cat no: T6522) and 0.1% BSA (Sigma, cat no. A9418) for 10 min. Afterwards, cells were pelleted at 14x g and lysed. cDNA was synthesized from total RNA using the RevertAid RT kit (Thermo Fisher, cat no: K1691) according to the manufacturer's protocol. SYBR Green qPCR SuperMix (Invitrogen, cat no: 11760100) and ViiA 7 Real-Time PCR System (Applied Biosystems) were used for qPCR analysis (qRT-PCR primers sequences are shown in Supplementary Table 6). All qPCR reactions were performed in duplicates. Data were generated and analyzed using QuantStudio™

Real-Time Real-Time PCR v1.1 software. All values were normalized to the level of *β-actin* or *Sdha* mRNA, as indicated in the figure legends.

### Culture and stimulation of patient-derived organoids (PDOs)

All patients enrolled in the study gave consent prior to PDO generation based on the institutional review board (IRB) project-number 207/15 of the Technical University Munich. Experimental procedures involving human subjects were performed in agreement with the ethical principles for medical research, as defined by the WMA Declaration of Helsinki and the Department of Health and Human Services Belmont Report. Patients have given their informed consent without being paid. The gender of the participants was determined on the basis of self-reports. No gender was excluded. All patients received standard surgical resection or fine-needle biopsy. Paraffin-embedded tissues were preprocessed by a pathologist after surgical resection or fine-needle biopsy and confirmed as PDAC prior to further investigation. PDOs were isolated as previously described[33] and cultured in floating collagen Type I matrices[58,59]. Briefly, a mixture of culture medium, cells, neutralizing solution (550 mM HEPES (Carl Roth, cat no: HN78.3) in 11x PBS) and collagen was incubated for 1 h at 37 °C until polymerization and the formation of a gel. Afterwards, 600 μL of media was added on top of the gel, which was loosen up with the help of a tip and let to float. For the first 72 h, the medium contained 3 mM Y-27632 (Biomol, cat no: Cay10005583), while after this time point, the medium was changed every 48 h. From day 7 onwards, PDOs were treated with 1 or 10 μM 5Z-7-Oxozeaenol (Sigma, cat no: O9890-1MG), while an equal volume of DMSO (Carl Roth, cat no: 7029.1) was added in the control gels. At day 9 of PDO development, organoids were fixed with 4% PFA (Alfa Aesar, cat no: J61899-AP) and further analyzed. The effect of 5Z-7-Oxozeaenol on PDOs was quantified using randomly captured pictures of living organoids at day 8 (24 h treatment) and fixed organoids at day 9 (48 h treatment). PDOs established from 3 different patients were analyzed. For bright field microscopy PDOs were imaged with a Leica DM IL LED microscope (Leica, Wetzlar, Germany) at 5x and 20x magnifications. The basic characteristics of the patient-derived organoid (PDO) lines are shown in Supplementary Table 2.

### Human subjects for PDAC-derived tumor spheroid and tumor tissue experiments

Human pancreatic tissues and serum were obtained from diseased patients undergoing surgery with informed consent from all patients for de-identified use at the Strasbourg University Hospitals, University of Strasbourg, France (DC-2016-2616 and RIPH2 LivMod IDRCB 2019-A00738-49, ClinicalTrial NCT04690972). The protocols were approved by the local Ethics Committee of the University of Strasbourg Hospitals ethical committee. All material was collected during a medical procedure strictly performed within the frame of the medical treatment of the patient. Informed consent is provided according to the Declaration of Helsinki. Detailed patient information and informed consent procedures are implemented by the Strasbourg University Hospital Biological Resources Center (HUS CRB). Patients were given an information sheet which outlines that their left-over biological material that was collected during their medical treatment is requested for research purposes. All patients received and signed an informed consent form without being paid (protocols DC-2016-2616 and RIPH2 LivMod IDRCB 2019-A00738-49 ClinicalTrial NCT04690972). The identity of the patients was protected by internal coding. Patient gender was determined on the basis of self-reports. No gender was excluded. The basic characteristics of the patient-derived spheroids are shown in Supplementary Table 3 and 5.

### Single-cell RNA-Seq on patient-derived tumor spheroids

Patient-derived tumor-spheroids were generated from patient adenocarcinoma pancreatic tissues undergoing surgical resection using a protocol published in ref. 35. Tissue was dissociated using

gentleMACS™ Octo Dissociator with Heaters and Human tumor dissociation kit (Miltenyi Biotec, cat no: 130-095-929) following manufacturer's instructions. Total cell populations including cancer cells, fibroblasts and immune cells was used to generate multicellular tumor-spheroids in Corning® 96-well Black/Clear Bottom Low Flange Ultra-Low Attachment Microplate (Corning). Cells were cultured in complete MammoCult™ Human Medium (Stemcell Technologies, cat no: 05620) supplemented with patient serum. After 24 h, tumor-spheroids were treated with 5Z-7-Oxozeaenol (25 μM, Sigma, cat no: O9890-1MG) or DMSO (Carl Roth, cat no: 7029.1) as a control overnight. Tumor-spheroids were harvested after treatment and dissociated using accutase. In total 76,652 cells were analyzed for DMSO control, and 78,467 cells for 5Z-7-Oxozeaenol-treatment. After cell washing using DPBS$^{-/-}$, antibody cell receptors were blocked using FcR Blocking Reagent (Miltenyi Biotec, cat no: 130-059-901), and CD45$^+$ cells were stained using CD45 Antibody, anti-human APC, REAfinity™ (1:50, Miltenyi Biotec, cat no: 130-110-633, clone: REA747, lot: 5220310224) or with the corresponding REA Control (S) APC antibody (1:50, Miltenyi Biotec, cat no: 130-110-434, clone: REA293, lot: 5220405741), according to manufacturer's instruction. Living cells were selected using Zombie green (BioLegend, cat no: 423111) staining according to manufacturer's instructions. CD45$^+$ cells were enriched by flow cytometry into 384 well cell capture plates (Single Cell Discovery, https://www.scdiscoveries.com) using SH800 cell sorter (Sony) as described[35]. Each well of a cell capture plate contains a small 50 nl droplet of barcoded primers and 10 μl of mineral oil (Sigma, cat no: M8410). Data were acquired using the Sony SH800 cell sorter software V2.1.5. Sorted plates were briefly centrifuged at 4 °C, snap-frozen on dry ice and stored at −80 °C until processed.

scRNA-Seq was performed by Single-Cell Discoveries B.V. using SORT-Seq, a modified CEL-Seq2 protocol[60]. Cells were heat-lysed at 65 °C followed by cDNA synthesis. After second-strand cDNA synthesis, all the barcoded material from one plate was pooled into one library and amplified using in vitro transcription (IVT). Following amplification, library preparation was done following the CEL-Seq2 protocol.3 to prepare a cDNA library for sequencing using TruSeq small RNA primers (Illumina). The DNA library was paired-end sequenced on an Illumina Nextseq™ 500, high output, with a 1×75 bp Illumina kit (read 1: 26 cycles, index read: 6 cycles, read 2: 60 cycles, Illumina, cat no: 20024906). During sequencing, Read 1 was assigned 26 base pairs and was used to identify the Illumina library barcode, cell barcode, and UMI. Read 2 was assigned 60 base pairs and used to map to the reference transcriptome Homo sapiens hg38 (including mitochondrial genes) with BWA-MEM. Data was demultiplexed as described in Grün et al.[61]. Mapping and generation of count tables were automated using the MapAndGo script. Unsupervised clustering and differential gene expression analysis was performed with the Seurat 3.2.2 R toolkit, as described[62]. For the comparison of cell clusters, variation of gene expression and of key pathways was determined by a NES obtained using Gene Set Enrichment Analysis (GSEA)[35,63,64]. Significance of the data was determined by the FDR values < 0.05. Data are available at NCBI Gene Expression Omnibus database (https://www.ncbi.nlm.nih.gov/geo/query/acc.cgi?acc=GSE223135) under accession number GSE223135.

## Single-cell RNA-Seq on patient-derived tumor tissue
Patient adenocarcinoma pancreatic tissues undergoing surgical resection was dissociated using gentleMACS™ Octo Dissociator with Heaters and Human Tumor dissociation kit (Miltenyi Biotec; cat no: 130-095-929) following manufacturer's instructions. The total cell fraction was stained using anti-human APC, REAfinity™ (1:50, Miltenyi Biotec, cat no: 130-110-633, clone: REA747, lot: 5220310224) and Zombie green viability kit (BioLegend, cat no: 423111) according to manufacturer's instruction. Living CD45$^+$ immune cells were separated

from other living cell populations using SH800 cell sorter (Sony). From the total cell population (437,234 total cells), 206,922 CD45+ living positive cells were isolated with 98.71% of sort efficiency. A post-sort control were performed by an analysis of the post-sort fraction, confirming the purity of the cell fraction. Data were acquired using the Sony SH800 cell sorter software V2.1.5. Isolated immune cells were cultured in 96 well plates in complete MammoCult™ Human Medium (Stemcell Technologies, cat no: 05620) supplemented with human proliferation supplement (3.4%, Stemcell Technologies, cat no: 05620), hydrocortisone (0.056%, Stemcell Technologies, cat no: 74142), heparin (0.011%, Stemcell, cat no: 07980), amphotericin B (Merck, cat no: A2942), primocin (InvivoGen, cat no: ant-pm-05) and patient serum. The other cell populations were cultured as tumor spheroids in Corning® 96-well Black/Clear Bottom Low Flange Ultra-Low Attachment Microplate in complete MammoCult™ Human Medium (Stemcell Technologies, cat no: 05620). After 3 days, tumor spheroids were treated using 5Z-7-Oxozeaenol (25 μM, Sigma, cat no: O9890-1MG) or DMSO (Carl Roth, cat no: 7029.1) as control. Medium was refreshed after 24 h to remove the compounds. After two more days, the conditioned media were used to stimulate immune cells. scRNA-Seq on the immune cell population was performed 24 h after stimulation. Living cells were sorted into 384 well cell capture plates (Single Cell Discovery, https://www.scdiscoveries.com) using SH800 cell sorter (Sony) as previously described[60]. Sorted plates were briefly centrifuged at 4 °C, snap-frozen on dry ice, and stored at −80 °C until processed.

scRNA-Seq library preparation was performed by Single-Cell Discoveries B.V. using SORT-Seq, a modified CEL-Seq2 protocol[62] (Single Cell Discovery, https://www.scdiscoveries.com). Reads were aligned to the human hG19 UCSC reference using Hisat2. R version 3.5.3 with Seurat and package "RaceID" for clusterization, cluster analysis and differentially expressed genes (DEG) calculation[35,62].

Data are available at NCBI Gene Expression Omnibus database (https://www.ncbi.nlm.nih.gov/geo/query/acc.cgi?acc=GSE275488) under accession number GSE275488.

## Perturbation studies in patient-derived tumor spheroids
Patient adenocarcinoma pancreatic tissues were processed as described above and cultured as tumor spheroids in complete MammoCult™ Human Medium (Stemcell Technologies, cat no: 05620). After cell aggregation, multicellular tumor spheroids were treated with 5Z-7-Oxozeaenol (1; 10 or 25 μM, Sigma, cat no: O9890-1MG) or DMSO (Carl Roth, cat no: 7029.1) as control. Viability was assessed after 3 days by measuring ATP levels using CellTiter-Glo® 3D Cell Viability Assay (Promega, cat no: G9681), according to the manufacturer's instructions.

## Histopathology and Immunohistochemistry (IHC)
Murine tissue and murine 3D collagen matrix samples were fixed in 4% paraformaldehyde and embedded in paraffin. Paraffin sections (2 μm) were stained with hematoxylin (Dako, cat no: CS700) and eosin (Sigma, cat no: HT110216) or various primary and secondary antibodies. Evaluation and quantification of ADM, PanIN and PDAC was performed by a pathologist based on the analysis of H&E sections.

For IHC in murine tissue and 3D collagen matrix samples, paraffin sections were deparaffinized with xylene and rehydrated in graded alcohols. Antigen retrieval was performed in EDTA-buffer (10 mM Tris (Carl Roth, cat no: 4855.2), 1 mM EDTA (Applichem, cat no: A4892.0500), 0.05% Tween (Carl Roth, cat no: 9127.1, pH 9) or citrate buffer (180 μM citric acid, Carl Roth, cat no: X863.1), 100 mM sodium citrate (Carl Roth, cat no: 3580.3)) at 98 °C for 30 min (or 5 min for 3D collagen matrix samples). Sections were treated with 3% hydrogen peroxidase (H$_2$O$_2$, Carl Roth, cat no: 8070.1) in water for 10 min to block endogenous peroxidase activity. Subsequently, unspecific

binding sites were blocked with normal horse serum (Vector Laboratories, cat no: S2000) for 30 min. IHC staining was performed with the following primary antibodies at 4 °C overnight: anti-α-Amylase (1:2000; Cell Signaling, cat no: 3796, clone: D55H10, lot: 3), anti-Keratin 17/19 (CK17/19) (1:1000, Cell Signaling, cat no: 3984, clone: D32D9, lot: 1), anti-SOX9 (1:2000, Merck Millipore, cat no: AB5535, lot: 2724407), anti-Ki-67 (1:2000, Thermo Scientific, cat no: RM-9106-S, clone: SP6, lot: 9106R0723), anti-phospho ERK (Thr202/Tyr204, 1:1000, Cell Signaling, cat no: 4370, clone: D13.14.4E, lot: 15), anti-cl. CASP3 (cl. CASP3) (1:750, Cell Signaling, cat no: 9661, lot: 47) and anti-RIPK3 (1:800, Enzo, cat no: ADI-905-242-100, lot: 03071904). Sections were then incubated with HRP-conjugated anti-rabbit IgG and were visualized with 3-3´-diamonobenzidine (DAB, Thermo Scientific, cat no: TA-125-QHDX). All sections were counterstained with hematoxylin (Dako, cat no: CS700). Image acquisition was performed on a Leica DM1000 microscope equipped with a Leica EC3 digital camera and quantification were performed as previously described[65].

Paraffin-embedded patient derived organoids (PDOs) were further processed for H&E staining. IHC in PDO samples was performed on a BondRxm automated stainer (Leica) with antibodies for anti-Ki-67 (1:50, Abcam, cat no: ab15580) and anti-cl. CASP3 (1:150, Cell Signaling, cat no: 9661, lot: 47). First, 2 μm slides were deparaffinized and pre-treated with epitope retrieval solution 1 (corresponding to citrate buffer pH 6) for 20 min, then incubated with primary antibodies diluted in antibody diluent (Leica, cat no: AR9352). Antibody binding was detected using the Polymer Refine detection kit (Leica, cat no: DS9800), without post-primary antibody and DAB (Thermo Scientific, cat no: TA-125-QHDX) as chromogen, and counterstained with hematoxylin (Dako, cat no: CS700). Finally, the slides were scanned with an automated slide scanner (Leica Biosystems), and the Aperio ImageScope software (Leica Biosystems) was used to take representative images.

The IHC staining of human pancreatic tissues included in the TMA was conducted as described above using anti-TAK1 antibody (1:2250, Novus Biologicals, cat no: NBP1-87819, A78535), anti-phospho TAK1 (Ser192, 1:300, BIOSS, cat no: bs-5435R, lot: AG07102355), anti-TAB3 (1:200, LS Bio, cat no: LS-B4705, lot: 31221). The staining was determined according to the widely used semi-quantitative Allred score, a sum of staining distribution and staining intensity (minimal 0, maximal 8 points)[66]. All commercial available antibodies were validated by the manufacturer.

## Immunoblot analysis

Protein lysates were prepared from pancreatic samples as described previously[65]. Protein extracts were separated by SDS-polyacrylamide gel electrophoresis (SDS-PAGE), transferred to PVDF membrane and analyzed by immunoblot with following antibodies (1:1000): anti-β-actin (Sigma, cat no. A2066, lot: 0000182447), anti-cl. CASP3 (cat no. 9661, lot: 47), anti-p-ERK (Thr202/Tyr204, cat no. 4377, clone: 197G2, lot: 12), anti-ERK (cat no. 4695, clone: 137F5, lot: 5), anti-p-Akt (Ser473, cat no. 4060, clone: D9E, lot: 27), anti-Akt (cat no. 4685, clone: 11E7, lot: 6), anti-p-MEK1/2 (Ser221, cat no. 2338, clone: 166F8, lot: 9), anti-JNK1/2 (cat no. 9258, clone: 56G8, lot: 11), anti-p-JNK1/2 (Thr183/Tyr185, cat no. 4668, clone: 81E11, lot: 2), anti-p38 (cat no. 9212, lot: 12), anti-p-p38 (Thr180/Tyr182, cat no. 9215, clone: 3D7, lot: 7), anti-I-κBα (cat no. 4812, clone: 44D4, lot: 13), anti-p-MLKL (Ser345, cat no. 37333, clone: D6E3G, lot: 2) (Cell Signaling), anti-RIPK3 (Novus, cat no. Img-5523-1, lot: 8337-1803), anti-MLKL (Merck Millipore, cat no. MABC604, clone: 3H1), anti-HSP90 (Enzo, cat no. ADI-SPA-830-D, clone: AC88, lot: 02011766) and anti-GAPDH (ABD Serotec, cat no. MCA4739, clone: 6C5, lot: 161665). As secondary antibodies, anti-rabbit-HRP (1:5000, Cytiva, cat no. NA934) and anti-mouse-HRP (1: 5000, Cytiva cat no. NA931) were used. Uncropped and unprocessed scans of all blots are available in the Source data file. All commercial available antibodies were validated by the manufacturer.

## RAS activation assay

RAS activity was assessed using RAS Activation Assay Kit (Merck Millipore, cat no: 17-218) according to the manufacturer's instructions. Briefly, RAS-GTP was pulled-down using an agarose-bound glutathione S-transferase (GST) fusion protein corresponding to the RAS binding domain (RBD) of RAF. RAS-GTP was detected by immunoblotting using an anti-RAS antibody (1:1000, Merck Millipore, cat no: 05-516, clone: RAS10). The antibody were validated by the manufacturer.

## Array comparative genomic hybridization (aCGH) analysis

To characterize copy number alterations, oligo array CGH using the Agilent platform (Boeblingen, Germany) was performed. Genomic DNA was extracted from formalin-fixed, paraffin-embedded (FFPE) tissue sections using the DNAeasy FFPE kit (Qiagen, cat no: 73504). DNA extracted from 4 normal pancreatic tissues was pooled and used as reference DNA. 250 ng of test and reference DNA were differentially labeled with Cy3-dUTP (test) and Cy5-dUTP (reference) by random primed labeling using the CGH labeling kit for oligo arrays (Enzo, cat no: ENZ-42671). Hybridization, washing and scanning was performed according to the manufacturer's protocol and the data were extracted from the Feature Extraction Software (Agilent) as tab-delimited text files. The arrays used were custom-designed 8x60k arrays (AMADID, cat no: 41078) with approx. 60,000 probes covering the whole mouse genome. The probe set includes the Agilent 44k (AMADID, cat no: 15028) in order to enable merging of the 60k data with 44k data. The raw data were imported into the R statistical platform (R Development Core Team. R: A language and environment for statistical computing. R Foundation for Statistical Computing, Vienna, Austria. ISBN 3-900051-07-0, URL: http://www.R-project.org/) and the background subtracted median intensity signals were used to build the log2 ratios. After median normalization, values were quality filtered using flags as defined by the Agilent Feature extraction software. The log2 ratios were subsequently segmented, called and copy number regions were defined using functions from the CGHbase[67], CGHcall[68] and CGHregions[69]. For the CGHcall function, 75% estimated proportion of tumor cells based on microscopic assessment of Haematoxylin stained FFPE sections was used. The copy number profiles were karyogram-style plotted using an in-house written R function. The aCGH data generated in this study have been deposited at NCBI Gene Expression Omnibus database https://www.ncbi.nlm.nih.gov/geo/query/acc.cgi?acc=GSE282891 under accession number GSE282891.

## Statistics

Data were analyzed by GraphPad Prism 9 built-in tests (specifics are indicated in figure legends). Data are presented relative to their respective controls. Results are expressed as the mean $n$ ± standard error of the mean (SEM). Statistical analyzes were performed for $n \geq 3$ samples, using parametric tests (one-way ANOVA) or non-parametric tests (two-tailed Mann-Whitney U test or Kruskal-Wallis test) as indicated in figure legends, after determination of distribution by the Shapiro-Wilk normality test. $p < 0.05$ were considered statistically significant. For scRNA-Seq analysis, transcriptome profile variations were determined by a NES obtained, using GSEA analysis. The significance of the data was determined by the FDR values. According to GSEA, results are significant if FDR < 0.05.

## Reporting summary

Further information on research design is available in the Nature Portfolio Reporting Summary linked to this article.

## Data availability

The aCGH data generated in this study have been deposited at NCBI Gene Expression Omnibus database https://www.ncbi.nlm.nih.gov/geo/query/acc.cgi?acc=GSE282891 under accession number GSE282891. The RNA-seq data generated in this study are available at

NCBI Gene Expression Omnibus database https://www.ncbi.nlm.nih.gov/geo/query/acc.cgi?acc=%20GSE223135 under the accession number GSE223135 or https://www.ncbi.nlm.nih.gov/geo/query/acc.cgi?acc=%20GSE275488 under accession number GSE275488. The remaining data are available within the Article, Supplementary Information or Source Data file. Source data are provided with this paper.

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

## Acknowledgements

The authors thank Dr. V. Dixit (Genentech, San Francisco, CA, USA) for kindly providing *Ripk3^-/-* mice and Dr. S. Akira (Osaka, Japan) for kindly providing *Tak1^Fl/Fl* mice, R. Hakem (Toronto, Canada) for kindly providing *Caspase-8^Fl/Fl mice*, CV. Wright (Nashville, USA) for kindly providing *Ptf1aCre* mice, T. Jacks and D. Tuveson (Massachusetts, USA) for kindly providing *B6.129-KRAS^tm4Tyj* mice. The authors are grateful for technical support from K. Wehr, L. Tenten (RWTH Aachen, Germany), N. Eichhorst, M. Suzanj, T. Janssen, M. Fastrich, V. Herbertz, C. Rupprecht, S. Lindner (HHU Düsseldorf, Germany) as well as C. Ponsolles and S. Durand (Inserm U1110, Strasbourg, France). We further thank Prof. Patrick Pessaux for providing PDAC patient tissues for single cell studies, Sarah Durand for managing the PDAC patient-derived tissue and Charlotte Bach and Clara Ponsolles for cell sorting. Furthermore, we are also grateful for the support and advice on cell sorting by Claudine Ebel and Muriel Phillips (IGBMC, flow cytometry platform, France) and thank Single Cell Discoveries for their single-cell sequencing services and data analysis. The HPDE cell line was generously provided by Dr. Anil K. Rustgi's laboratory (New York, USA), to whom we are very grateful. Work in the laboratory of T.L. was funded by the German Cancer Aid (Deutsche Krebshilfe – 70114893), the European Research Council (ERC) under the European Union's Horizon 2020 research and innovation program through the ERC Consolidator Grant PhaseControl (Grant Agreement 771083) as well as the MODS project funded from the programme "Profilbildung 2020" (grant no. PROFILNRW-2020-107-A), an initiative of the Ministry of Culture and Science of the State of North Rhine-Westphalia. The laboratory of T. L. was further supported by the German Research Foundation (DFG - LU 1360/3-2 (279874820), LU 1360/4-1 (461704932), CA 830/3-1 (440603844, co-application with M.C.), 493659010 and SFB-CRC 1382 Project A01), the German Ministry of Health (BMG – DEEP LIVER 2520DAT111) and from the medical faculty of the Heinrich Heine University. M.V. was supported by the German Cancer Aid (Deutsche Krebshilfe – 70114893) and the research committee of the Medical Faculty of Heinrich Heine University (2021-58). A.S. was funded by the German Research Foundation (DFG, German Research Foundation – SCHN 1659/1-1, 457724578) and the research committee of the Medical Faculty of the Heinrich Heine University (2022-23). T.Lu., M.V., and A.S. received funding from the Ministry of Culture and Science of the State of North Rhine-Westphalia (CANTAR – NW21-062E). V.K. was funded by the research committee of the Medical Faculty of the Heinrich Heine University (2023-32). M.H. was supported by the German-Research-Foundation (SFB-TR179 and SFB-TR209) and an ERC Consolidator grant (LiverHepatoMetabopath), the I&I future Helmholtz Topic, the EOS Flundern grant and a MOST grant. M.H. has received funding in this project from the European Union's Horizon 2020 research and innovation programme under grant agreement No 667273.

M.R. and A.P. acknowledge financial support from the German Cancer Aid (Max Eder Program, Deutsche Krebshilfe 111273, M.R.) and the German Research Foundation (DFG, SFB1321 Modeling and Targeting Pancreatic Cancer, Projects S01 and TP12, Project ID 329628492 and 424465722 and 531385338 to M.R.). M.R. is supported by the Bavarian Ministry of Economic Affairs, EISglobe (LSM-2104- 0017), BMBF projects SATURN3 (01KD2206P), QuE-MRT (13N16450) and FAIRPACT (01KD2208B) and DKTK (German Cancer Consortium) Strategic Initiative Organoid Platform. T.F.B. was supported by the European Research Council Grant ERC-AdG-2020 FIBCAN, ARC Grant TheraHCC2.0 IHUARC IHU201301187, French National Research Agency LABEX ANR-10-LABX-0028_HEPSYS (T.F.B.) and the Institute Universitaire de France (IUF). M.R. acknowledge the financial support by the German Cancer Aid (Max-Eder Program 111273 and 70114328) and the German Research Foundation (DFG, SFB1321 Modeling and Targeting Pancreatic Cancer, Projects S01 and TP12, Project ID329628492 and Project RE 3723/4-1).

## Author contributions

T.L., M.V., C.K., and A.T.S. designed the project. C.K., A.T.S., M.T.S., E.C., A.P., K.S., F.B., A.Z., M.S., V.L., V.B., L.K., M.C., V.K., and T.N. performed experiments. C.K., A.T.S., M.T.S., E.C., M.V., F.J., J.M., F.B., H.E., K.U., V.K., M.M.G., N.G., M.C., M.H., and M.R. analyzed and interpreted data. T.L., S.M., A.G., J.G.H., M.R., M.L., R.M.S., J.N.K., H.A., R.R., M.H., T.H., T.F.B., W.T.K., I.E., and F.B. provided resources and critical input. C.K., A.T.S., V.K., M.V., and T.L. wrote the manuscript with input from all authors. A.T.S. and C.K. contributed equally to this work.

## Funding

## Competing interests

The authors declare no competing interests.

## Additional information

Anne T. Schneider[1], Christiane Koppe[1], Emilie Crouchet[2], Aristeidis Papargyriou [3,4,5], Michael T. Singer[1], Veronika Büttner[1], Leonie Keysberg[1], Marta Szydlowska [6], Frank Jühling[2], Julien Moehlin[2], Min-Chun Chen [4], Valentina Leone[3,6,7], Sebastian Mueller [8,9], Thorsten Neuß [10], Mirco Castoldi [1], Marina Lesina[11], Frank Bergmann[12,13], Thilo Hackert [14,15], Katja Steiger [16], Wolfram T. Knoefel[17], Alex Zaufel[1], Jakob N. Kather [18,19,20], Irene Esposito[21], Matthias M. Gaida[22,23,24,25], Ahmed Ghallab [26,27], Jan G. Hengstler [26], Henrik Einwächter[4], Kristian Unger[28,29], Hana Algül [11], Nikolaus Gassler[30], Roland M. Schmid[4], Roland Rad [9,31], Thomas F. Baumert [2,32,33,34], Maximilian Reichert[3,4,35,36,37], Mathias Heikenwalder [6,38], Vangelis Kondylis [1,40], Mihael Vucur [1,40] & Tom Luedde [1,39,40] ✉

[1]Department of Gastroenterology, Hepatology and Infectious Diseases, University Hospital Düsseldorf, Medical Faculty at Heinrich-Heine-University, Duesseldorf, Germany. [2]University of Strasbourg, Inserm, Institute for Translational Medicine and Liver Disease (ITM), UMR_S1110, Strasbourg, France. [3]Translational Pancreatic Cancer Research Center, Klinik und Poliklinik für Innere Medizin II, Klinikum rechts der Isar, Technical University of Munich, Munich, Germany. [4]Klinik und Poliklinik für Innere Medizin II, Klinikum rechts der Isar, Technical University of Munich, Munich, Germany. [5]Institute of Stem Cell Research, Helmholtz Center Munich, German Research Center for Environmental Health, Neuherberg, Germany. [6]Division of Chronic Inflammation and Cancer, German Cancer Research Center (DKFZ), Heidelberg, Germany. [7]Research Unit Radiation Cytogenetics, Helmholtz-Zentrum München, German Research Center for Environmental Health, Neuherberg, Germany. [8]Institute of Molecular Oncology and Functional Genomics, School of Medicine, TU Munich, Munich, Germany. [9]Center for Translational Cancer Research (TranslaTUM), School of Medicine, Technical University of Munich, Munich, Germany. [10]Lehrstuhl für Biophysik E27, Center for Protein Assemblies (CPA), Technical University Munich (TUM), Garching, Germany. [11]Comprehensive Cancer Center München, Institute for Tumor Metabolism, TUM School of Medicine and Health, Technical University of Munich, Munich, Germany. [12]Institut of Pathology, Heidelberg University Hospital, Heidelberg, Germany. [13]Clinical Pathology, Klinikum Darmstadt GmbH, Darmstadt, Germany. [14]Department of General, Visceral, and Transplantation Surgery, University Hospital Heidelberg, Heidelberg, Germany. [15]Department of General, Visceral and Thoracic Surgery, University Hospital Hamburg-Eppendorf, Hamburg, Germany. [16]Institute of Pathology, School of Medicine, Technical University of Munich, Munich, Germany. [17]Department of Surgery A, Heinrich-Heine-University Düsseldorf and University Hospital Düsseldorf, Duesseldorf, Germany. [18]Else Kroener Fresenius Center for Digital Health (EFFZ), Technical University Dresden, Dresden, Germany. [19]Division of Pathology and Data Analytics, Leeds Institute of Medical Research at St James's, University of Leeds, Leeds, UK. [20]Medical Oncology, National Center for

Tumor Diseases (NCT), University Hospital Heidelberg, Heidelberg, Germany. [21]Institute of Pathology, University Hospital Duesseldorf, Heinrich-Heine University, Duesseldorf, Germany. [22]Institute of Pathology, University Medical Center Mainz, JGU-Mainz, Mainz, Germany. [23]Research Center for Immunotherapy, University Medical Center Mainz, JGU-Mainz, Mainz, Germany. [24]Joint Unit Immunopathology, Institute of Pathology, University Medical Center, JGU-Mainz, Mainz, Germany. [25]TRON, Translational Oncology at the University Medical Center, JGU-Mainz, Mainz, Germany. [26]Leibniz Research Centre for Working Environment and Human Factors (IfADo) at the Technical University Dortmund, Dortmund, Germany. [27]Forensic Medicine and Toxicology Department, Faculty of Veterinary Medicine, South Valley University, Qena, Egypt. [28]Department of Radiation Oncology, University Hospital, LMU Munich, Munich, Germany. [29]Research Unit Translational Metabolic Oncology, Institute for Diabetes and Cancer, Helmholtz Zentrum München Deutsches Forschungszentrum für Gesundheit und Umwelt (GmbH), Neuherberg, Germany. [30]Section Pathology of the Institute of Forensic Medicine, University Hospital Jena, Jena, Germany. [31]Department of Internal Medicine II, Klinikum Rechts der Isar, Technical University of Munich, Munich, Germany. [32]Pôle des Pathologies Hépatiques et Digestives, Service d'Hepato-Gastroenterologie, Strasbourg University Hospitals, Strasbourg, France. [33]Institut Hospitalo-Universitaire (IHU) Strasbourg, Strasbourg, France. [34]Institut Universitaire de France (IUF), Paris, France. [35]Center for Organoid Systems (COS), Technical University of Munich, Garching, Germany. [36]Munich Institute of Biomedical Engineering (MIBE), Technical University of Munich, Garching, Germany. [37]German Center for Translational Cancer Research (DKTK), Munich, Germany. [38]The M3 Research Institute, Karls Eberhards Universität Tübingen, Tübingen, Germany. [39]Center for Integrated Oncology Aachen Bonn Cologne Düsseldorf (CIO ABCD), Düsseldorf, Germany. [40]These authors contributed equally: Vangelis Kondylis, Mihael Vucur, Tom Luedde. ✉e-mail: luedde@hhu.de

