## [Transparent Peer Review file · Nature Communications]

A decision point between transdifferentiation and programmed cell death priming controls KRAS-dependent pancreatic cancer development

Corresponding Author: Professor Tom Luedde

Version 0:

Reviewer comments:

Reviewer #1

(Remarks to the Author)

Schneider et al. performed in vivo and in vitro analyses of TAK1-mediated transdifferentiation and regulation of programmed cell death in KRAS-dependent pancreatic cancer development using genetically engineered mice and other models. In particular, the involvement of TAK1 was analyzed in detail using knockout mice and inhibitors, and a therapeutic strategy for human pancreatic cancer by TAK1 inhibition was proposed.

Overall, the experiments have proceeded steadily and have shown reliable results, but the following points regarding TAK1 involvement need to be experimentally demonstrated.

1. Although it is important evidence that TAK1 expression is upregulated in PDAC, data that TAK1 activation is actually occurring have not been presented. Data on the expression status of activated TAK1 and phosphorylated TAK1 are also required to be presented. In addition, the importance of TAK1 activation in pathogenesis can be demonstrated by examining the expression of TAB1 and TAB2/3, which are necessary for TAK1 activation.
2. TAK1 is one of the MAP3Ks, and there are other MAPKs downstream besides NF- κ B. In this study, the authors consider the involvement of ERKs as well as KRAS, but TAK1 rather has an important function in the activation of JNK and p38, which are stress response MAPKs. Therefore, evidence on the activation status of those stress response kinases is required.
3. Data using TAK1 knockout mice are reliable. However, the concentration of 5Z-7-oxozeaenol used as TAK1 inhibitor seems to be somewhat high. Many previous studies have used 1 μ M or less, but this study used 10 μ M and in some cases 25 μ M. It is necessary to clearly indicate the reason for this concentration setting and to mention the results at the concentrations that are usually commonly used. This is important because this paper proposes that TAK1 inhibition can lead to a therapeutic strategy.
4. Recently, KRAS-K12D inhibitors such as MRTX1133 have been clinically proven to have therapeutic efficacy against PDAC. In this context, it will be interesting to examine the effects of KRAS-G12D inhibitors on TAK1 activation, transdifferentiation, and cell death.

Reviewer #2

(Remarks to the Author)

The authors have presented an intriguing discovery demonstrating that genetic ablation or pharmacological inhibition of TAK1 prevents pancreatic cancer development by inducing programmed cell death (both apoptosis and necroptosis) in KRAS mutant transdifferentiated cancer cells during acinar-to-ductal metaplasia (ADM). In addition to cell culture assays, the authors have corroborated the TAK1 inhibition-induced elimination of tumor cells in patient-derived organoids. Notably, there were no evident injury-associated inflammatory responses resulting from this treatment. This finding provides a fresh perspective on the potential of TAK1 inhibition in anti-cancer therapy.

I do have some minor concerns regarding certain aspects of the data, which the author should take into consideration:

1. In Figure 1, the authors have demonstrated that the IKK inhibitor TPCA-1 did not impede acinar cell transdifferentiation, leading them to assert that TAK1's function is not attributed to NF- κ B inhibition but rather through alternative downstream pathways. This conclusion may be somewhat premature. The authors should employ multiple cell lines and NF- κ B inhibitors

- to validate this point. Furthermore, it would be beneficial to include a discussion on the potential downstream pathways.
2. In Figure 2, with regard to the upregulation of death genes in KRAS-mutant mice but not in TAK1 deletion mice, the authors should address whether KRAS mutation itself caused this or if it is simply a consequence of TAK1 inhibition-induced cell death that eliminates these cells with high death gene expression.
 3. In Figure 5, aside from cleaved caspase-3, did the author also detect signals for p-MLKL?
 4. In Figure 1e, the last word of the headline appears to be incomplete. Please ensure that it is properly presented.
 5. The authors should delve into the rationale behind the absence of pro-inflammatory responses following TAK1 inhibition (although necroptosis happened, why no pro-inflammatory molecules released), as this holds significant potential benefits for patients.

Reviewer #3

(Remarks to the Author)

In this conceptually interesting manuscript, Schneider et al. describe a crosstalk between transforming growth factor β -activated kinase 1 (TAK1) and receptor-interacting protein kinase 1 (RIPK1) as a decision point between transdifferentiation and priming to programmed cell death in KRAS-dependent development of pancreatic cancer.

A main issue of the current version is that there is not enough information provided on the antibodies that were used that allows reproduction of results and that for several figures only qualitative data is shown and no statistical significance is provided.

Figures 1e and 1f need to show a larger area of the 3D culture, showing more transdifferentiation events. A quantification for $n=3$ independently conducted repeats needs to be included for these figures. In both figures the brightfield pictures are of poor quality.

Figures 2b and 2c need quantification analysis of $n=3$ independently conducted repeats for each staining. In Figure 2c the staining for cleaved caspase 3 is not very convincing. Why are the DMSO treated ductal structures show cleaved caspase 3 positivity? In Figure 2f a larger area should be shown, since the IHC staining is a bit weak.

The inhibition of necroptosis and apoptosis restores metaplasia of TAK1 deficient cells (Figure 3), indicating that TAK1 is not a driver of ADM. What is the explanation for this? Doesn't this counteract the model in Figure 6? What is the NF-kappaB activity status in these cells?

The antibodies used for IHC and Western blotting are not described sufficiently such that they will allow reproducing the results. Ordering numbers and identification of phosphorylation site for which the authors probed should be included.

Some of the figure legends are not accurate. For example, Figure 2f does not show $n=4$ samples as stated.

Reviewer #4

(Remarks to the Author)

The manuscript entitled "A decision point between transdifferentiation and programmed cell death priming controls KRAS-dependent pancreatic cancer development" by Schneider et al. underscores the pivotal role of TAK1 in the ADM process. The manuscript suggests that inhibiting this enzyme has the potential to not only inhibit ADM but also impede the growth of patient-derived PDAC organoids. This hypothesis is rooted in the premise that TAK1 regulates cell survival through the phosphorylation of the IKK complex catalytic subunits and the suppression of programmed cell death (PCD) by inhibiting RIPK1 activity. The authors present compelling evidence for the successful inhibition of this enzyme, resulting in the suppression of acinar cell transdifferentiation in genetic mouse models and organoids. Furthermore, the manuscript highlights that TAK1 serves as a critical junction point between cell survival and PCD during ADM, ultimately inducing cell death in tumor cells within PDAC patient-derived organoids, thus offering a promising strategy for the treatment of established PDAC.

While the research is methodologically robust and well-executed, the findings do not seem to introduce substantially novel insights or transformative concepts that would significantly advance our understanding of this field.

Specific comments

1. The main claim in the manuscript revolves around the importance of TAK1 in the development of KRAS-dependent pancreatic cancer. While the authors demonstrate the ability to reduce cancer cell viability in two human PDAC cell lines (as mentioned on page 12), it would be beneficial to explore whether this effect, if it exists, would be less pronounced in KRAS wild-type PDAC cell lines. Therefore, I recommend the authors consider repeating their experiments using KRAS wild-type cell lines.

2. The utilization of the TAK1 inhibitor has implications not only in a chemopreventive context but also as an anti-tumor strategy. However, there is a concern that the TAK1 inhibitor may yield different outcomes between early-stage and late-stage tumors, as previously discussed in PMID: 37625401. It would be valuable to clarify the stage or progression level of the organoids when they were treated. Additionally, considering the implementation of an orthotopic mouse model, as suggested in PMID: 37625401, may help assess the differences in drug efficacy between early- and late-stage tumors.

3. The first subtitle, "TAK1 deficiency in ADM and PDAC development," in the results section on page 6, does not fully represent the data. It is recommended to revise the subtitle to accurately reflect the data presented.

4. Extended figure 2b illustrates the phosphorylation of downstream targets of KRAS signaling, but it does not include the TAK1 Δ AC control, unlike extended figure 2a. Is there a specific reason why the results of 18 weeks old TAK1 Δ AC are not

included in extended figure 2b?

5. The authors employed 5Z-7-oxozeaenol to inhibit TAK1, using a specific concentration (10uM). It would be beneficial for the authors to explain the rationale behind choosing this particular concentration and whether increasing the concentration could have additional effects.

6. In Figure 4, the authors claim that there is no 'obvious difference in the chromosomal aberrations and the spectrum of affected tumor genes between KRASG12D single transgenic mice and KRASG12D TAK1/CASP8ΔAC RIPK3^{-/-} mice.' However, there appear to be differences in several PDAC-associated genes (e.g., Tgfbr2). To address this, a quantitative comparison is warranted to clarify the extent of these differences.

7. In Figure 5d-f, the authors state that these figures depict significant transcriptomic changes in activated CD3⁺CD8⁺ cells between non-treatment and treatment. It is recommended to provide quantitative measurements of the significant changes and specify what percentage of cells in each cluster exhibited differences between non-treatment and treatment.

8. The exclusion of the NFκB pathway from the mechanism behind TAK1 was demonstrated by the treatment of an IKK inhibitor (TPCA-1) in vitro. However, this exclusion was not validated in vivo. To strengthen this assertion, it would be prudent to validate the exclusion of the NFκB pathway in vivo as well, as the current exclusion may be considered somewhat premature without in vivo validation.

9. Figure 5g is not mentioned in the text, while Figure 5h is referenced in the text but appears to be absent. It is advisable to clarify whether Figure 5g was mistakenly referenced as Figure 5h, or if there was an oversight in the presentation of these figures.

Version 1:

Reviewer comments:

Reviewer #1

(Remarks to the Author)

Thank you for your various additional experiments in response to my comments and for your convincing results. I believe that these results provide stronger evidence for the involvement of TAK1 and have improved the overall quality of the paper. Therefore, the revised manuscript is ready for acceptance.

Reviewer #2

(Remarks to the Author)

The authors have done a good job in the revised manuscript. My questions have been adequately addressed.

Reviewer #3

(Remarks to the Author)

My points have been addressed sufficiently.

Reviewer #4

(Remarks to the Author)

The authors have addressed the previously raised issues,

Point-by-Point Response to the Referees:

Reviewer #1 (Remarks to the Author):

Schneider et al. performed in vivo and in vitro analyses of TAK1-mediated transdifferentiation and regulation of programmed cell death in KRAS-dependent pancreatic cancer development using genetically engineered mice and other models. In particular, the involvement of TAK1 was analyzed in detail using knockout mice and inhibitors, and a therapeutic strategy for human pancreatic cancer by TAK1 inhibition was proposed. Overall, the experiments have proceeded steadily and have shown reliable results, but the following points regarding TAK1 involvement need to be experimentally demonstrated.

First of all, we would like to thank reviewer #1 for the fair review and constructive comments on our manuscript that helped us clarify the TAK1-dependent mechanism underlying the observed phenotype through additional experiments during the revision process. Below are our answers to the specific queries raised by the reviewer.

1. Although it is important evidence that TAK1 expression is upregulated in PDAC, data that TAK1 activation is actually occurring have not been presented. Data on the expression status of activated TAK1 and phosphorylated TAK1 are also required to be presented. In addition, the importance of TAK1 activation in pathogenesis can be demonstrated by examining the expression of TAB1 and TAB2/3, which are necessary for TAK1 activation.

We are grateful to reviewer #1 for this important comment. Indeed, TAK1 expression is not an proof of its activation. As requested by the reviewer, we have performed additional IHC stainings for TAB1, TAB2 and TAB3, as well as for the phosphorylated, active form of TAK1.

The new data showed a similar expression profile of TAB3 in PDAC tissue. Similarly, strong p-TAK1 signal was also found in PDAC specimen, as it was the case for total TAK1. Unfortunately, we were unable to establish a reliable staining for TAB1 and TAB2. Together, these data indicate that, in addition to its increased expression, TAK1 is activated in human PDAC tissue (**data set 1**).

Data set 1 / new Fig. 1a, b and Supplementary Fig. 1: Expression of TAB3 and p-TAK1 in human PDAC tissue.

2. *TAK1* is one of the MAP3Ks, and there are other MAPKs downstream besides NF- κ B. In this study, the authors consider the involvement of ERKs as well as KRAS, but *TAK1* rather has an important function in the activation of JNK and p38, which are stress response MAPKs. Therefore, evidence on the activation status of those stress response kinases is required.

As requested by the referee, we additionally examined the levels of p-JNK, JNK, p-p38, p38 in lysates of pancreatic tissue from 6-week and 18-week-old mice. As shown in **data set 2 / new Supplementary Fig. 3a, b**, immunoblotting analysis showed that neither JNK nor p38 activation was altered in pancreatic lysates of mice with the indicated genotypes at both time points, suggesting no major involvement for these stress kinases.

Data set 2 / new Supplementary Fig. 3a, b: Immunoblot analyses of p-JNK, JNK, p-p38 and p38 in lysates of pancreatic tissue from 6-week and 18-week-old mice.

3. Data using *TAK1* knockout mice are reliable. However, the concentration of 5Z-7-oxozeaenol used as *TAK1* inhibitor seems to be somewhat high. Many previous studies have used 1 μ M or less, but this study used 10 μ M and in some cases 25 μ M. It is necessary to clearly indicate the reason for this concentration setting and to mention the results at the concentrations that are usually commonly used. This is important because this paper proposes that *TAK1* inhibition can lead to a therapeutic strategy.

We thank the reviewer for this important comment. The relatively high concentrations of 10 μ M and 25 μ M 5Z-7-oxozeaenol were originally selected, because the inhibitor was used as a single death-inducing stimulus, while in the literature it is typically used at 1 μ M but with the addition of TNF, which would certainly have been sufficient to achieve a stronger effect in our experimental settings (see also our response to the comment 1 from reviewer #4). However, as requested by the reviewer, we investigated further this issue in the revised version (see **data set 3**).

Similar to literature reports, we now show that treatment of three PDAC cell lines with 10 μ M 5Z-7-Oxozeaenol for 48h only led to a marginal cell death induction, but combined treatment

with 2 μM 5Z-7-Oxozeanol and 20 ng/ml TNF induced significantly stronger cell death (**data set 3A**). Furthermore, we performed comparative analyses with 1, 10 and 25 μM 5Z-7-oxozeanol in all critical experiments using organoids and spheroids. In murine organoids formed $\text{KRAS}^{\text{G12D}}$ -expressing pancreatic acinar cell explants in 3D collagen matrix cultures, we observed a dose-dependent effect of 5Z-7-oxozeanol, with 1 μM having a milder effect than 10 μM , which led to a complete disintegration of the duct-like structures (**data set 3B**). Similarly, 5Z-7-Oxozeanol also led to disintegration of PDAC patient-derived organoids (PDOs) in a concentration- and time-dependent manner (**data set 3C**). Furthermore, a clear concentration-dependent reduction in cell viability was also observed the patient-derived spheroids with 25 μM 5Z-7-Oxozeanol displaying the strongest effect (**data set 3D**). Finally, it is unlikely that the effect of 10 μM 5Z-7-Oxozeanol would be the result of unspecific toxicity, as the use of Nec-1s alone or in combination with *Z-VAD-fmk* prevented the disintegration of murine organoids, suggesting that TAK1 inhibition induced RIPK1 kinase-activity dependent cell death (see **Fig. 2e**).

Taken together, these data indicate that $\text{KRAS}^{\text{G12D}}$ -expressing acinar cells or organoids are sensitive to TAK1 inhibition in a concentration-dependent manner, and these data are now included in the revised version.

Data set 3/ new Fig. 4b, Fig. 5i, Supplementary Fig. 2g, Supplementary Fig. 5 d,e: Dose-dependent effect of TAK1-inhibition on patient derived organoids and murine primary acinar cells. **A)** Cell death evaluation in human pancreatic cancer cell lines treated with DMSO (solvent), TNF (20 ng/ml), 5Z-7-Oxozeaenol (10 μ M) or TNF (20 ng/ml)/5Z-7-Oxozeaenol (2 μ M) for 48 h. Cell death was assessed by measuring the released LDH-to-total LDH ratio. Results are expressed as mean \pm SEM. n=3. **B)** KRAS^{G12D}-expressing pancreatic acinar cell explants were treated 24h after duct formation with 5Z-7-oxozeaenol (1 and 10 μ M) for additional 48h. n=3. **C)** Patient derived organoids were either treated with 1 μ M or 10 μ M 5Z-7-oxozeaenol for 48h (A total of n= 21, n = 23 and n = 33 PDOs (24h) and n = 108, n = 90 and n = 97 PDOs (48h) treated with DMSO, 1 μ M 5Z-7-Oxozeaenol and 10 μ M 5Z-7-Oxozeaenol were analyzed, respectively). **D)** PDAC-derived tumour spheroids were subjected to treatment with 5Z-7-oxozeaenol (1; 10 or 25 μ M). Viability was assessed after 3 days by measuring ATP levels. The graph shows mean \pm of RLU (relative light unit n=6 per group). * p < 0.05; ** p < 0.005; *** p < 0.001; **** p < 0.0001. Ordinary one-way ANOVA followed by Tukey's multiple comparisons test or Kruskal-Wallis test with Dunn's multiple-comparisons test, respectively. Red arrows indicate duct-like structures. Red arrows indicate duct-like structures.

4. Recently, KRAS-K12D inhibitors such as MRTX1133 have been clinically proven to have therapeutic efficacy against PDAC. In this context, it will be interesting to examine the effects of KRAS-G12D inhibitors on TAK1 activation, transdifferentiation, and cell death.

We would like to thank this reviewer for this interesting suggestion. In line with previous findings (PMID: 36472553; 38294344), we found that MRTX1133 inhibited ERK phosphorylation in PANC-1 cells that bear a KRAS^{G12D} mutation, while it had only a minor effect on MIA-PaCa2 cells that have a KRAS^{G12C} mutation (**data set 4A**).

Next, we addressed the effect of MRTX1133 on our *in vitro* ADM assay assessing the transdifferentiation and formation of duct-like structures from acinar explants of KRAS^{G12D} mice. Treatment with increasing concentrations of MRTX1133 (200 nM and 400 nM) for 3 days showed a dose-dependent inhibition of duct-like structure formation (**data set 4B**), which was not an effect of cytotoxicity, as the inhibitor did not lead to elevated cell death (**data set 4C**). In line with our RNA sequencing results reported in our original manuscript, we confirmed by qRT-PCR the transcriptional upregulation of a group of cell death-associated molecules (*Ripk3*, *Mkl1*, *Casp8*, *Casp3*) but not of *Ripk1* and *Casp9* (**data set 4D**). Interestingly, the expression of these upregulated genes was still increased upon MRTX1133 treatment, suggesting that the upregulation was not related to KRAS activation and duct-like formation. Moreover, we tested the expression of markers of acinar and ductal cell identity and found a strong downregulation of the acinar markers *Amylase 1*, *Amylase 2* and *Mist-1*, while the ductal marker *Ck19* was markedly upregulated. Again, KRAS^{G12D} inhibitor did not appear to change the expression of these cell fate markers. Moreover, the bona fide NF- κ B target gene *Tnfrsf25* (A20) was also significantly upregulated and its expression was unaffected by MRTX1133 (**data set 4D**), implicating that activation of the TAK1/NF- κ B signalling pathway during this ADM assay is independent of KRAS^{G12D} activity.

Collectively, inhibition of KRAS^{G12D} by MRTX1133 inhibited duct-like structure formation in our *in vitro* ADM assay but it did not affect TAK1/NF- κ B activation, the transcriptional upregulation of cell death genes, and cell death induction.

Data set 4 / new Supplementary Fig. 4b-4e: The effect of the MRTX1133 on TAK1 activation, transdifferentiation, and cell death in KRAS^{G12D} expressing pancreatic acinar cells. **A**) Immunoblotting analysis for pERK in extracts of control- and MRTX1133-treated (50 nM, 200 nM, 1 µM) MIA-PaCa2 and PANC-1 cells (24h) showing the effective inhibition on KRAS^{G12D} activity induced by MRTX1133. **B-C**) ADM rate of KRAS^{G12D} acinar cell explants upon 3-day culture on collagen matrix. MRTX1133 treatment successfully prevented duct-like structure formation without inducing cytotoxicity. n=3. **D**) qRT-PCR for cell death genes (*Ripk1*, *Ripk3*, *Casp-3*, *Casp-8*, *Casp-9*, *Mikl*), cell fate makers (*Amy1*, *Amy2*, *CK19*, *Mist1*) and NF-κB target gene (*Tnfaip3*) under the indicated treatment. n=3. * p < 0.05; ** p < 0.005; *** p < 0.001; Ordinary one-way ANOVA followed by Tukey's multiple comparisons test.

Reviewer #2 (Remarks to the Author):

The authors have presented an intriguing discovery demonstrating that genetic ablation or pharmacological inhibition of TAK1 prevents pancreatic cancer development by inducing programmed cell death (both apoptosis and necroptosis) in KRAS mutant transdifferentiated cancer cells during acinar-to-ductal metaplasia (ADM). In addition to cell culture assays, the authors have corroborated the TAK1 inhibition-induced elimination of tumor cells in patient-derived organoids. Notably, there were no evident injury-associated inflammatory responses resulting from this treatment. This finding provides a fresh perspective on the potential of TAK1 inhibition in anti-cancer therapy.

I do have some minor concerns regarding certain aspects of the data, which the author should take into consideration:

We thank the reviewer and appreciate his/her assessment of the potential of TAK1 as an anti-cancer therapeutic agent. We have now addressed the questions raised by Reviewer #2 point-by-point below.

1. In Figure 1, the authors have demonstrated that the IKK inhibitor TPCA-1 did not impede acinar cell transdifferentiation, leading them to assert that TAK1's function is not attributed to NF- κ B inhibition but rather through alternative downstream pathways. This conclusion may be somewhat premature. The authors should employ multiple cell lines and NF- κ B inhibitors to validate this point. Furthermore, it would be beneficial to include a discussion on the potential downstream pathways.

In order to further investigate this important aspect in the best possible way, we were able to demonstrate the effect of NF- κ B inhibition *in vivo* by gaining access to a dataset of a previously performed experiment (PMID: 27454298) in which the NF- κ B subunit RelA/p65 was deleted in the very same KRAS^{G12D} mice (**data set 5**). Importantly and in clear contrast to the TAK1 deficiency, the additional deletion of *Rela* did not prevent transdifferentiation and PDAC formation in KRAS^{G12D} mice. A careful pathological evaluation even indicated an exacerbation of PDAC development upon *Rela* ablation. This opposing effect of acinar cell-specific *Tak1* vs. *Rela* deficiency further supports our conclusion that TAK1 mediates pancreatic oncogenesis through a function independent of activating NF- κ B signalling pathway. As requested by the referee, we have expanded our discussion on this important point in the revised manuscript (page 17).

Data set 5 / new Supplementary Fig. 2l, m: RelA is not necessary for KRAS^{G12D} dependent PDAC development. **A)** Haematoxylin and eosin (H&E) staining in pancreatic tissue sections from the indicated KRAS^{G12D} and KRAS^{G12D} RelA^{ΔAc} mice aged 46 to 61 weeks. **B)** Quantification of healthy pancreas tissue (Normal), acinar-to-ductal metaplasia (ADM), pancreatic intraepithelial neoplasia 1 (PanIN-1) and PanIN-2 in the indicated animals (n = 5 for KRAS^{G12D} mice and n = 8 for KRAS^{G12D} RelA^{ΔAc} mice). The KRAS^{G12D} mice share the same genetic background.

2. In Figure 2, with regard to the upregulation of death genes in KRAS-mutant mice but not in TAK1 deletion mice, the authors should address whether KRAS mutation itself caused this or if it is simply a consequence of TAK1 inhibition-induced cell death that eliminates these cells with high death gene expression.

We believe that the reviewer refers to the Western blot analysis in Figure 2a, where the expression and activation of MLKL (p-MLKL) and cl. caspase-3 in TNF-treated primary acinar cells is analysed. The impression that total MLKL decreased in TAK1-deficient KRAS^{G12D} cells, as compared to TAK1-proficient cells, is due to the slightly unequal loading, as evidenced by the expression of the housekeeper gene β-Actin. We have now performed a densitometric quantification of the expression levels, which shows no differences in total MLKL levels between the 3 different genotypes.

Data set 6: Densitometric analysis of immunoblot signals in lysates of pancreatic acinar cells isolated from the indicated mice.

3. In Figure 5, aside from cleaved caspase-3, did the author also detect signals for p-MLKL?

The immunohistochemical detection of MLKL activation is a general challenge with the commercially available antibodies (see the paper of Samson and colleagues, PMID: 33589776). Unfortunately, we have not been able to establish a p-MLKL staining that was sufficiently specific to allow a solid conclusion.

4. In Figure 1e, the last word of the headline appears to be incomplete. Please ensure that it is properly presented.

We thank the reviewer for pointing out this mistake. The headline is now presented properly.

5. The authors should delve into the rationale behind the absence of pro-inflammatory responses following TAK1 inhibition (although necroptosis happened, why no pro-inflammatory molecules released), as this holds significant potential benefits for patients.

The reviewer raised here a valid concern that we also found interesting, namely the lack of pro-inflammatory responses from patient spheroids treated with TAK1 inhibitor. Because in our original manuscript the entire spheroid culture, including the immune cells, had been exposed to TAK1 inhibitor, we could not exclude that the lack of inflammation was the result of NF- κ B signalling impairment in the immune cells. To avoid the concomitant inhibition of TAK1 in immune cells by 5Z-7-oxozeaenol, we performed a new, conceptually-changed spheroid experiment as follows: The PDAC cells were sorted and separated in CD45⁺ immune cells and CD45⁻ cells e.g. cancer cells, fibroblasts etc. The immune cell-depleted spheroids were then treated overnight with 5Z-7-Oxozeaenol or DMSO, followed by removal of the inhibitor and culture in 5Z-7-Oxozeaenol-free medium for 48h to obtain conditioned medium containing factors released by the dying cancer cells. The conditioned medium was used to stimulate the sorted immune cells, which were subsequently processed for scRNA-Seq analysis.

Interestingly, this modified experimental protocol confirmed the lack of activated immune response (**data set 7**). In detail, a total of 956 immune cells was analysed (DMSO: n = 529 cells; TAK1i: n = 427 cells). Similar clustering and transcriptomic profiles between control and treated condition indicate no effect of TAK1i-conditioned medium on immune cell population. This conclusion was supported by a GSEA pathway analysis showing no significant difference between the condition in all the identified clusters (**data set 7**). This result is in line with a previous observation in hepatocarcinogenesis, where we were able to show that the reactivity of necroptotic death was linked to competence for NF- κ B signalling activation in the dying cells. In this liver cancer model, we have shown that necroptotic hepatocytes with impaired NF- κ B signalling (as is now the case with TAK1-inhibited PDAC tumour cells) undergo a hyporeactive cell death that was not associated with a pro-inflammatory or pro-carcinogenic immune response PMID: **37329888**. However, when NF- κ B was activated in the necroptotic hepatocytes, the cell death response was pro-inflammatory and pro-carcinogenic. We have discussed this aspect in detail in the revised manuscript and we would like to thank reviewer #2 for his/her valuable comments that helped us to improve the characterisation of the role/function of NF- κ B respectively cell death and significantly improve the manuscript.

Data set 7 / new Fig. 5h-k, Supplementary Fig. 5g.h and Supplementary Data 2-4: Single cell RNA-Seq analysis showed no effect of TAK1i-conditioned medium on immune cell responses. **A)** Experimental setup for the generation of tumour spheroids and for the inhibition of TAK1 with subsequent stimulation of immune cells. PDAC tumour tissue was dissociated and the cells were sorted. CD45⁺ immune cells were isolated from the whole cell population. Other cell types were used to generate tumour spheroids. After 3 days, tumour spheroids were treated with TAKi (25 μ M) or DMSO as a control. The medium was renewed after 24 hours to remove the compounds. After two more days, the conditioned medium was collected and used to stimulate immune cells. 24 hours after stimulation, scRNA-Seq analysis of the immune cell population was performed. **B)** Analysis of spheroid cell viability followed TAKi-treatment based on ATP levels (RLU = relative light unit) (mean \pm sd) 48h after refreshing the medium. **C)** 2D-visualization of single-cell transcriptomics using UMAPs are shown. A total of 956 immune cells was analysed (DMSO: n = 529 cells; TAKi: n = 427 cells). **D)** Similar clustering and transcriptomic profiles between control and treated condition indicate no effect of conditioned medium on immune cell populations. **E)** Summary of GSEA pathway analyses from distinct inflammatory pathways that was found significantly dysregulated in the T cells cluster I in the first spheroid experiment.

Reviewer #3 (Remarks to the Author):

In this conceptually interesting manuscript, Schneider et al. describe a crosstalk between transforming growth factor β -activated kinase 1 (TAK1) and receptor-interacting protein kinase 1 (RIPK1) as a decision point between transdifferentiation and priming to programmed cell death in KRAS-dependent development of pancreatic cancer. A main issue of the current version is that there is not enough information provided on the antibodies that were used that allows reproduction of results and that for several figures only qualitative data is shown and no statistical significance is provided. Figures 1e and 1f need to show a larger area of the 3D culture, showing more transdifferentiation events. A quantification for $n=3$ independently conducted repeats needs to be included for these figures. In both figures the brightfield pictures are of poor quality. Figures 2b and 2c need quantification analysis of $n=3$ independently conducted repeats for each staining.

We thank reviewer #3 for raising these important points, which we have addressed in the revised version of the manuscript. Regarding the antibodies used in this study, detailed information is provided within the methods part. Moreover, as requested by the reviewer, we performed a statistical examination for old Figures 1e, 1f, 2b and 2c and we have included new pictures. The results are presented in **data sets 8 and 9** below. In summary, the quantification confirms the original statement that the deletion or pharmacological inhibition of TAK1 prevents the formation of KRAS^{G12D}-induced ductal structures (old Fig. 1e and 1f / **data set 8**) or mediates their cell death (old Fig. 2b and 2c/ **data set 9**). In the revised version of the manuscript, we have combined old figure 2b and 2c, as they were redundant.

Data set 8 / new Fig. 1f, g, i, j: **A)** Bright-field images, H&E staining and IHC of SOX9 in pancreatic acinar explants from the indicated mice grown in 3D collagen matrix culture. Results from pancreatic explants isolated from 3 mice per genotype are presented. **B)** Bright-field images, H&E staining and IHC of SOX9 of pancreatic acinar explants isolated from a KRAS^{G12D} mouse and grown in 3D cell culture with indicated supplements - untreated, DMSO (solvent), 5Z-7-oxozeaenol (10 μ M) or TPCA-1 (10 μ M) (left panel). Quantification of duct-like structures based on the bright-field images. n=3 independent animals. *** p < 0.001, **** p < 0.0001; n.s.= non-significant; Ordinary one-way ANOVA followed by Tukey's multiple comparisons test. The red arrows indicate duct-like structures, the green arrows indicate SOX9 positive duct-like structures.

B
Data set 9 / new Fig. 2d, e: A) Bright-field images, H&E staining and IHC of SOX9, Ki-67 and cl. Casp-3 in KRAS^{G12D}-expressing pancreatic acinar cell explants during transdifferentiation in 3D collagen matrix culture. Treatments were performed on pancreatic explants isolated from 3 mice. The medium was supplemented with DMSO, 5Z-7-oxozeaenol (10 μ M) alone or in combination with Z-VAD-FMK (10 μ M) and/or Nec-1s (20 μ M) for 24h when the first signs of transdifferentiation occurred. Quantification of duct-like structures was performed on bright-field images. ** $p < 0.005$, n.s.= non-significant; Ordinary one-way ANOVA followed by Tukey's multiple comparisons test. **B)** Additional examples of IHC of SOX9, Ki-67 and cl. Casp-3 in KRAS^{G12D}-expressing pancreatic acinar cell explants during transdifferentiation in 3D collagen matrix culture.

The different arrows indicate positive cells (yellow (cl. Casp3), green (Sox9), pink (Ki67)) or duct-like structures on bright-field-pictures (red).

In Figure 2c the staining for cleaved caspase 3 is not very convincing. Why are the DMSO treated ductal structures show cleaved caspase 3 positivity? In Figure 2f a larger area should be shown, since the IHC staining is a bit weak.

We would like to thank the reviewer for this comment, as we have obviously not indicated the labelling in the Figure clearly enough. The DMSO-treated structure shows no staining for cleaved caspase 3, but is positive for Sox9, as expected (**data set 10A**). We have adjusted the Figure and as suggested by the reviewer, we have included larger areas of the RIPK3 staining both of transdifferentiated duct-like structures and mouse pancreatic tissue sections in the **data set 10B-C**.

Data set 10 / new Fig. 2d, h, i: H&E staining and IHC of pancreatic acinar cell explants and pancreatic tissue sections. A) Bright-field images, H&E staining and IHC of SOX9, Ki-67 and cl. Casp-3 in KRAS^{G12D}-expressing pancreatic acinar cell explants during transdifferentiation in 3D collagen matrix culture. Treatments were performed on pancreatic explants isolated from 3 mice. The medium was supplemented with DMSO, 5Z-7-oxozeanol (10 µM) alone or in combination with Z-VAD-FMK (10 µM) and/or Nec-1s (20 µM) for 24h when the first signs of transdifferentiation occurred. B) IHC of RIPK3 in KRAS^{G12D}-expressing pancreatic acinar cell explants in 3D cell culture during transdifferentiation. C) RIPK3 IHC in pancreatic tissue sections from the indicated mice at 18 weeks of age. The red arrows indicate duct-like structures confirmed on bright-field microscopy, green arrows indicate Sox9 positive duct-like structures, pink arrows indicate Ki67 positive cells and the yellow arrows cl. Casp-3 positive cells.

The inhibition of necroptosis and apoptosis restores metaplasia of TAK1 deficient cells (Figure 3), indicating that TAK1 is not a driver of ADM. What is the explanation for this? Doesn't this counteract the model in Figure 6? What is the NF-kappaB activity status in these cells?

We fully agree with the reviewer's conclusion that TAK1 is not the driver of ADM. Rather, next to its important function in activating NF- κ B, TAK1 also represents a cell death checkpoint that controls the threshold for cell death initiation during transdifferentiation. Thus, deletion or inhibition of TAK1 facilitates cell death induction, thereby preventing ADM and tumour development. Accordingly, the additional combined inhibition of apoptosis and necroptosis restored ADM and tumour development further supporting this hypothesis. Therefore, in our model in Figure 6, we do not imply that TAK1 drives/triggers ADM and PDAC, as KRAS activation does. Our results suggest that functional TAK1 support oncogenesis by suppressing death of transdifferentiating cells.

As requested by the reviewer, we examined the NF- κ B activity status by IHC. As expected, no nuclear signal for RelA/p65 could be observed by IHC on pancreatic sections of mice lacking TAK1 (**data set 11A**). To further investigate the specific role of NF- κ B in mediating ADM and PDAC development, we gained access to a dataset of a previously performed experiment (PMID: 27454298) in which the NF- κ B subunit RelA/p65 was deleted in the very same KRAS^{G12D} mice (**data set 11B**). Importantly and in clear contrast to the TAK1-mutants, the additional deletion of *Rela* did not prevent transdifferentiation and PDAC formation in KRAS^{G12D} mice. A careful pathological evaluation even indicated an exacerbation of PDAC development upon *Rela* ablation. This opposing effect of acinar cell-specific TAK1 vs. RelA deficiency further supports our conclusion that TAK1 mediates pancreatic oncogenesis through a function independent of activating NF- κ B signalling pathway.

Data set 11 / new Supplementary Fig. 2l, m: NF-κB dependence in KRAS^{G12D} mediated ADM and PDAC. A) IHC for RelA/p65 shows no nuclear localisation suggesting that NF-κB activation is inhibited in the indicated mice, as expected due to the TAK1 deletion. B) Haematoxylin and eosin (H&E) staining in pancreatic tissue sections from KRAS^{G12D} and KRAS^{G12D} RelA^{ΔAc} mice aged 46-61 weeks. Quantification of healthy pancreas tissue (Normal), acinar-to-ductal metaplasia (ADM), pancreatic intraepithelial neoplasia 1 (PanIN-1) and PanIN-2 in the indicated animals (n=5 for KRAS^{G12D} mice and n=8 for KRAS^{G12D} RelA^{ΔAc} mice). The KRAS^{G12D} mice share the same genetic background.

The antibodies used for IHC and Western blotting are not described sufficiently such that they will allow reproducing the results. Ordering numbers and identification of phosphorylation site for which the authors probed should be included.

This has been amended in our revised version. All antibody order numbers, their target site and the dilutions that were used are included in the figures or the methods part.

Some of the figure legends are not accurate. For example, Figure 2f does not show n=4 samples as stated.

We thank the reviewer for this important comment. We have now checked and amended all figures legends that were inaccurate. Old Figure 2f shows representative RIPK3 staining from mice with the indicated genotypes. Sections from 4 different mice per genotype were examined, therefore n=4.

Reviewer #4 (Remarks to the Author):

The manuscript entitled "A decision point between transdifferentiation and programmed cell death priming controls KRAS-dependent pancreatic cancer development" by Schneider et al. underscores the pivotal role of TAK1 in the ADM process. The manuscript suggests that inhibiting this enzyme has the potential to not only inhibit ADM but also impede the growth of patient-derived PDAC organoids. This hypothesis is rooted in the premise that TAK1 regulates cell survival through the phosphorylation of the IKK complex catalytic subunits and the suppression of programmed cell death (PCD) by inhibiting RIPK1 activity. The authors present compelling evidence for the successful inhibition of this enzyme, resulting in the suppression of acinar cell transdifferentiation in genetic mouse models and organoids. Furthermore, the manuscript highlights that TAK1 serves as a critical junction point between cell survival and PCD during ADM, ultimately inducing cell death in tumor cells within PDAC patient-derived organoids, thus offering a promising strategy for the treatment of established PDAC. While the research is methodologically robust and well-executed, the findings do not seem to introduce substantially novel insights or transformative concepts that would significantly advance our understanding of this field.

We thank the reviewer for his/her positive assessment of the quality of our manuscript. Prompted by the reviewers' specific queries, we were able to further assess the function of TAK1 inhibition, the dependency on the KRAS status, as well as the contribution of NF- κ B and cell death in the context of KRAS-dependent transdifferentiation and pancreatic cancer development. This has allowed us to reveal new aspects that were not previously considered. Our responses to the individual queries of this reviewer are shown below.

Specific comments

1. The main claim in the manuscript revolves around the importance of TAK1 in the development of KRAS-dependent pancreatic cancer. While the authors demonstrate the ability to reduce cancer cell viability in two human PDAC cell lines (as mentioned on page 12), it would be beneficial to explore whether this effect, if it exists, would be less pronounced in KRAS wild-type PDAC cell lines. Therefore, I recommend the authors consider repeating their experiments using KRAS wild-type cell lines.

We thank the reviewer for raising this important point. We have now explored in more detail the role of KRAS status in conferring susceptibility to TAK1 inhibition using 3 different pancreatic cancer cell lines (BxPC3 > KRAS^{WT}; HPAC > KRAS^{G12D}; MIA-PaCa2 > KRAS^{G12C}) (**data set 12**). The cell death sensitivity of these 3 PDAC cell lines was assessed by measuring the ratio of released-to-total LDH upon treatment with 5Z-7-oxozeaenol alone or in combination with TNF for 48 hours. Our analyses revealed that treatment with 10 μ M 5Z-7-Oxozeaenol led only to a marginal cell death induction in all three PDAC cell lines. However, combined treatment of the cells with 20 ng/ml TNF and 2 μ M 5Z-7-Oxozeaenol induced significantly stronger cell death no matter whether the cells were bearing WT or mutant KRAS (**data set 12A**). Of note, TNF stimulation alone did not lead to increased cell death in all 3 cell lines, highlighting the additional effect of TAK1 inhibition, namely lowering the threshold of cells to death. These results suggested that human PDAC cell lines, independent of their KRAS status, are rather resistant to TAK1 inhibition alone even at a relatively high concentration likely due to the increased mutational burden accumulated over the years.

In addition, we have examined the cell death susceptibility of immortalized human pancreatic ductal epithelial cells (HPDE) upon Dox-inducible expression of GFP or KRAS^{G12D}. Dox-mediated KRAS^{G12D} induction for 3 days led to the development of characteristic cell vacuolization and the upregulation of phosphorylated ERK (**data set 12B**). Subsequent incubation of HPDEs with 5Z-7-Oxozeaenol alone or in combination with TNF led to a significant induction of cell death independent of KRAS^{G12D} expression (**data set 12B**). Interestingly, HPDEs (with or without KRAS^{G12D} expression) showed higher sensitivity to 5Z-7-oxozeaenol alone compared to the PDAC cell lines, further implying that transformed PDAC cells acquire resistance to cell death.

Altogether, these results show that KRAS mutational status in PDAC cancer cells is not intrinsically linked to a differential susceptibility to cell death upon treatment with 5Z-7-oxozeaenol. Both KRAS WT and mutated cells are sensitive to TAK1 inhibition either alone or in combination with TNF, suggesting that TAK1 inhibitor could be an effective killing approach in a KRAS-independent manner.

Data set 12 / new Supplementary Fig. 4f-4h and 5d-f: Cell death induction by TAK1 inhibition in KRAS mutated and WT pancreatic cancer cell lines: BxPC3 > KRAS^{WT}; HPAC > KRAS^{G12D}; MIA-PaCa2 > KRAS^{G12C}. **A)** Cell death was determined by the ratio of released-to-total LDH (lactate dehydrogenase) 48h after stimulation. **C:** DMSO: control group, TAK1i: 5Z-7-oxozeaenol alone (10 μ M), TNF (20 ng/ml), TAK1i and TNF: TNF (20 ng/ml), 5Z-7-oxozeaenol (2 μ M). n=3-4. **B)** Representative bright-field microscopy images of the immortalized human pancreatic duct epithelial cells (HPDEs) treated with 400 ng/ml doxycycline to induce the expression of human KRAS^{G12D} or

GFP. Immunoblotting analysis of p-ERK^{Thr202/Tyr204} and β -Actin (loading control) in lysates of HPDE cells treated with 400 ng/ml doxycycline to induce the expression of human KRAS^{G12D} or GFP for the indicated time. Cell death evaluation in HPDEs treated with 400 ng/ml doxycycline for three days to induce KRAS^{G12D} or GFP expression, followed by incubation with DMSO (solvent), TNF (20 ng/ml), 5Z-7-Oxozeaenol (10 μ M) or (20 ng/ml)/5Z-7-Oxozeaenol (10 μ M) for additional 48 h. Cell death was assessed by measuring the released LDH-to-total LDH ratio. Results are expressed as mean \pm SEM. Each data point represents the result of each individual experiment (n = 3). Each treatment per experiment was performed in duplicates. Statistics were done by one-way ANOVA followed by Tukey's multiple comparisons test. * p < 0.05; ** p < 0.005; **** p < 0.0001; ns = not significant.

2. The utilization of the TAK1 inhibitor has implications not only in a chemopreventive context but also as an anti-tumor strategy. However, there is a concern that the TAK1 inhibitor may yield different outcomes between early-stage and late-stage tumors, as previously discussed in PMID: 37625401. It would be valuable to clarify the stage or progression level of the organoids when they were treated. Additionally, considering the implementation of an orthotopic mouse model, as suggested in PMID: 37625401, may help assess the differences in drug efficacy between early- and late-stage tumors.

We thank the reviewer for this important comment. We agree with the reviewer that TAK1 inhibition may have different effects in early- vs. late-stage tumours, as discussed in the Cancer Cell paper (PMID: **37625401**). Indeed, our *in vivo* genetic data using the Cre/loxP system only address the role of TAK1 inhibition during the early stages of pancreatic oncogenesis. Due to the regulatory requirements and restrictions for animal experiments to which we are subject to, we were unable to conduct *in vivo* experiments that would address the role of TAK1 inhibition in established tumours, as these described in the study indicated by the reviewer. To do this, we would have had to submit a new application (pilot study) for the required mouse experiments, as we have not established the relevant models in our lab. Therefore, it would not have been possible to apply for and carry out these experiments within the allocated revision period.

Instead, we have tried to simulate the effect of TAK1 inhibition on developing and established tumours by performing *in vitro* experiments on primary mouse and PDAC patient organoids. By applying the TAK1 inhibitor during the transdifferentiation and duct-like formation/ADM of mouse acinar explants, we have modelled the effect on the early-stage development. On the other hand, application of the inhibitor in already formed mouse organoids or human organoids and spheroids provides indications of the potential efficacy of TAK1 inhibition in late-stage pancreatic tumours. In particular, the fact that TAK1 inhibitor kills patient-derived organoids/spheroids that originate from established PDAC tumour tissues and already bear multiple oncogenic mutations underscores a clear beneficial potential for treating advanced PDACs. We have included further information on the staging and grading of the organoids in the revised version (**Supplementary Table 2 or see below**). The patient-derived organoids originate from advanced PDAC tumours.

Supplementary Table 2. Basic characteristics of the patient-derived organoid (PDO) lines

Patient-derived organoid (PDO) lines	KRAS status (Sanger-Seq)	Source	Gender	Age at the time of diagnosis	p Staging	Grading
Patient 1	G12D	Surgery	male	60-70	ypT3ypN1(3/23)Pn1	-
Patient 2	G12A	FNB	female	60-70	-	-
Patient 3	G12D	Surgery	male	70-80	pT3pN1(1/35)Pn1	G2-3

FNB: fine-needle biopsy

Altogether, we believe that our results suggest that TAK1 might be a promising target for treatment of both early (ADM/PanIN) and late (PDAC) tumours, but obviously, this will need to be further evaluated in future studies in preclinical models, as the reviewer suggests.

3. The first subtitle, "TAK1 deficiency in ADM and PDAC development," in the results section on page 6, does not fully represent the data. It is recommended to revise the subtitle to accurately reflect the data presented.

We thank the reviewer for this suggestion. We have now revised the subtitle as follows: TAK1 deficiency suppresses KRAS-driven ADM and PDAC development.

4. Extended figure 2b illustrates the phosphorylation of downstream targets of KRAS signaling, but it does not include the TAK1 Δ AC control, unlike extended figure 2a. Is there a specific reason why the results of 18 weeks old TAK1 Δ AC are not included in extended figure 2b?

The TAK1 Δ AC control was originally included in the 6-week-old mice to demonstrate that the single KO has no effect on the signalling pathways, and therefore, it was not considered necessary to be added in the immunoblot of the 18-week-old samples. However, as suggested by the reviewer, we have now added the TAK1 Δ AC control to the immunoblotting analysis of the 18-week-old samples (**data set 13**). Here it can also be seen that the TAK1 Δ AC pancreatic tissue lysates do not exhibit activation of the different signalling pathways analyzed. In line with the point 2 from reviewer #1, we extended our analyses to the stress kinases p38 and JNK, which are now also part of the new Supplementary Fig. 3a, b. Both stress kinases were not dysregulated in all examined genotypes.

Data set 13/ new Supplementary Fig. 3a, b: Western blot analyses in lysates of pancreatic tissue from 6-week and 18-week-old mice.

5. The authors employed 5Z-7-oxozeaenol to inhibit TAK1, using a specific concentration (10 μ M). It would be beneficial for the authors to explain the rationale behind choosing this particular concentration and whether increasing the concentration could have additional effects.

We thank the reviewer for this important comment, which was also raised by Reviewer 1. The relatively high concentrations of 10 μ M and 25 μ M 5Z-7-oxozeaenol were originally selected, because the inhibitor was used as a single death-inducing stimulus, while in the literature it is typically used at 1 μ M but with the addition of TNF, which would certainly have been sufficient to achieve a stronger effect in our experimental settings. However, we investigated further this issue in the revised version (see **data set 14**).

Similar to literature reports, we now show that treatment of three PDAC cell lines with 10 μ M 5Z-7-Oxozeaenol for 48h only led to a marginal cell death induction, but combined treatment with 2 μ M 5Z-7-Oxozeaenol and 20 ng/ml TNF induced significantly stronger cell death (**data set 14A**). Furthermore, we performed comparative analyses with 1, 10 and 25 μ M 5Z-7-oxozeaenol in all critical experiments using organoids and spheroids. In murine organoids formed KRAS^{G12D}-expressing pancreatic acinar cell explants in 3D collagen matrix cultures, we observed a dose-dependent effect of 5Z-7-oxozeaenol, with 1 μ M having a milder effect than 10 μ M, which led to a complete disintegration of the duct-like structures (**data set 14B**). Similarly, 5Z-7-Oxozeaenol also led to disintegration of PDAC patient-derived organoids (PDOs) in a concentration- and time-dependent manner (**data set 14C**). Furthermore, a clear concentration-dependent reduction in cell viability was also observed the patient-derived spheroids with 25 μ M 5Z-7-Oxozeaenol displaying the strongest effect (**data set 14D**). Finally, it is unlikely that the effect of 10 μ M 5Z-7-Oxozeaenol would be the result of unspecific toxicity, as the use of Nec-1s alone or in combination with Z-VAD-*fmk* prevented the

disintegration of murine organoids, suggesting that TAK1 inhibition induced RIPK1 kinase-activity dependent cell death (see Fig. 2e).

Taken together, these data indicate that KRAS^{G12D}-expressing acinar cells or organoids are sensitive to TAK1 inhibition in a concentration-dependent manner, and these data are now included in the revised version.

Data set 14 / new Fig. 4c, Supplementary Fig. 6a, Supplementary Fig. 2g: Dose-dependent effect of TAK1-inhibition on patient derived organoids and murine primary acinar cells. A) A) Cell death evaluation in human pancreatic cancer cell lines treated with DMSO (solvent), TNF (20 ng/ml), 5Z-7-Oxozeanol (10 μM) or TNF (20 ng/ml)/5Z-7-Oxozeanol (2 μM) for 48 h. Cell death was assessed by measuring the released LDH-to-total LDH ratio. Results are expressed as mean ± SEM. n=3. B) KRAS^{G12D}-expressing pancreatic acinar cell explants were treated 24h after duct formation with 5Z-7-oxozeanol (TAKi) (1 and 10 μM) for additional 48h. n=3. C) Patient derived organoids were either treated with 1 μM or 10 μM 5Z-7-oxozeanol for 48h (A total of n= 21, n = 23 and n = 33 PDOs (24h) and n = 108, n = 90 and n = 97 PDOs (48h) treated with DMSO, 1 μM 5Z-7-Oxozeanol and 10 μM 5Z-7-Oxozeanol were analyzed, respectively). D) PDAC-derived tumour spheroids were subjected to treatment with 5Z-7-oxozeanol (1; 10 or 25 μM). Viability was assessed after 3 days by measuring ATP levels. The graph shows mean ± of RLU (relative light unit n=6 per group). * p < 0.05; ** p < 0.005; * p < 0.001; **** p < 0.0001. Ordinary one-way ANOVA followed by Tukey's multiple**

comparisons test or Kruskal-Wallis test with Dunn's multiple-comparisons test, respectively. Red arrows indicate duct-like structures. Red arrows indicate duct-like structures.

6. In Figure 4, the authors claim that there is no 'obvious difference in the chromosomal aberrations and the spectrum of affected tumor genes between *KRAS^{G12D}* single transgenic mice and *KRAS^{G12D} TAK1/CASP8 Δ Ac RIPK3^{-/-}* mice.' However, there appear to be differences in several PDAC-associated genes (e.g., *Tgfbr2*). To address this, a quantitative comparison is warranted to clarify the extent of these differences.

In the analysis, chromosomal aberrations were detected in both genetic configurations, and, in particular, alterations in regions of genes that are known to have functions, for example, as co-mutations in pancreatic cancer, were highlighted. Based on the reviewer's comment, we have revised our statement. From our perspective, and also limited by the small sample size, there is no clear pattern of alterations that is consistently found in the majority of samples/tumors in either genetic background. Therefore, we wanted to avoid overinterpreting our findings. The revised statement now reflects this view (**data set 15**).

Data set 15 / Fig. 3j: Map of pancreatic cancer associated genes located in the chromosomal gain and loss regions detected by aCGH analysis in old Figure 4a. Genes were retrieved from the Cancer Gene Census. Columns next to each chromosome represent individual mice. Genotypes are labelled by horizontal-coloured bars. Mice with PDAC are labelled with a green star.

7. In Figure 5d-f, the authors state that these figures depict significant transcriptomic changes in activated *CD3⁺CD8⁺* cells between non-treatment and treatment. It is recommended to provide quantitative measurements of the significant changes and specify what percentage of cells in each cluster exhibited differences between non-treatment and treatment.

As requested by the reviewer, we now report the p-value and FDR q-value for the significant transcriptomic changes in the T cells from cluster 1 reported in the GSEA analysis (new **Fig. 5k**, new **Supplementary Data 1** or see below).

GeneSet	Nominal p-value	FDR q-value
GOBP_REGULATION_OF_RESPONSE_TO_STRESS	0.0	0.0
GOBP_LYMPHOCYTE_CHEMOTAXIS	0.0020242915	0.008892668
GOMF_C_C_CHEMOKINE_BINDING	0.0	0.0006526106
GOBP_IMMUNE_RESPONSE	0.0	0.00048324207
GOMF_CYTOKINE_RECEPTOR_ACTIVITY	0.0	0.0
GOMF_IMMUNE_RECEPTOR_ACTIVITY	0.0	0.0
REACTOME_CELLULAR_RESPONSES_TO_STIMULI	0.0	0.0
REACTOME_CYTOKINE_SIGNALING_IN_IMMUNE_SYSTEM	0.0	0.0011187063
REACTOME_INTERFERON_SIGNALING	0.0	0.00005804973
REACTOME_CHEMOKINE_RECEPTORS_BIND_CHEMOKINES	0.0	0.0

Further, we also provide the T cell distribution within each identified cluster based on the expression of specific markers and the treatment (DMSO vs. 5Z-7-oxozeaenol) in new **Supplementary Table 3**.

Supplementary Table 3. T lymphocyte populations per cluster detected in PDAC patient-derived tumor-spheroids

T lymphocyte populations	Cluster_0			Cluster_1			Cluster_2		
	total amount (n)	DMSO-treated (%)	5Z-7-Oxozaenol-treated (%)	total amount (n)	DMSO-treated (%)	5Z-7-Oxozaenol-treated (%)	total amount (n)	DMSO-treated (%)	5Z-7-Oxozaenol-treated (%)
CD3E	75	86,7 (n = 65)	13,3 (n = 10)	42	81,0 (n = 34)	19,0 (n = 8)	22	81,8 (n = 18)	18,2 (n = 4)
CD4	21	90,5 (n = 19)	9,5 (n = 2)	8	100,0 (n = 8)	0,0 (n = 0)	4	100,0 (n = 4)	0,0 (n = 0)
CD8A	39	100,0 (n = 39)	0,0 (n = 0)	16	87,5 (n = 14)	12,5 (n = 2)	19	73,7 (n = 14)	26,3 (n = 5)
CD44	75	88,0 (n = 66)	12,0 (n = 9)	47	78,7 (n = 37)	21,3 (n = 10)	23	82,6 (n = 19)	17,4 (n = 4)

Moreover, we have adapted the original experimental design in response to the request of reviewer #2 comment 5 that questioned the observed lack of pro-inflammatory responses. Because in our original manuscript the entire spheroid culture, including the immune cells, had been exposed to TAK1 inhibitor, we could not exclude that the lack of inflammation was the result of NF- κ B signalling impairment in the immune cells. Based on this concern and to avoid the concomitant inhibition of TAK1 in immune cells by 5Z-7-oxozeaenol, we performed a new, conceptually-changed spheroid experiment as follows: The PDAC cells were sorted and separated in CD45⁺ immune cells and CD45⁻ cells e.g. cancer cells, fibroblasts etc. The immune cell-depleted spheroids were then treated overnight with 5Z-7-Oxozaenol or DMSO, followed by removal of the inhibitor and culture in 5Z-7-Oxozaenol-free medium for 48h to obtain conditioned medium containing factors released by the dying cancer cells. The conditioned medium was used to stimulate the sorted immune cells, which were subsequently processed for scRNA-Seq analysis.

Interestingly, this modified experimental protocol confirmed the lack of activated immune response (**data set 16**). In detail, a total of 956 immune cells was analysed (DMSO: n = 529 cells; TAK1i: n = 427 cells). Similar clustering and transcriptomic profiles between control and treated condition indicate no effect of TAK1i-conditioned medium on immune cell population. This conclusion was supported by a GSEA pathway analysis showing no significant difference between the condition in all the identified clusters (**data set 16**). This result is in line with a previous observation in hepatocarcinogenesis, where we were able to show that the reactivity of necroptotic death was linked to competence for NF- κ B signalling activation in the dying cells. In this liver cancer model, we have shown that necroptotic hepatocytes with impaired NF- κ B signalling (as is now the case with TAK1-inhibited PDAC tumour cells) undergo a hyporeactive cell death that was not associated with a pro-inflammatory or pro-carcinogenic immune response PMID: **37329888**. However, when NF- κ B was activated in the necroptotic hepatocytes, the cell death response was pro-inflammatory and pro-carcinogenic. We have discussed this aspect in detail in the Discussion section of the revised manuscript.

Data set 16 / new Fig. 5h-k, Supplementary Fig. 5m: Single cell RNA-Seq analysis showed no effect of TAK1i-conditioned medium on immune cell responses. **A)** Experimental setup for the generation of tumour spheroids and for the inhibition of TAK1 with subsequent stimulation of immune cells. PDAC tumour tissue was dissociated and the cells were sorted. CD45+ immune cells were isolated from the whole cell population. Other cell types were used to generate tumour spheroids. After 3 days, tumour spheroids were treated with 5Z-7-oxozeaenol (25 μ M) or DMSO as a control. The medium was renewed after 24 hours to remove the compounds. After two more days, the conditioned medium was collected and used to stimulate immune cells. 24 hours after stimulation, scRNA-Seq. analysis of the immune cell population was performed. **B)** Analysis of spheroid cell viability followed TAKi-treatment based on ATP levels (RLU = relative light unit) (mean \pm sd) 48h after refreshing the medium. **C)** 2D-visualization of single-cell transcriptomics using UMAPs are shown. A total of 956 immune cells was analysed (DMSO: n = 529 cells; TAKi: n = 427 cells). **D)** Similar clustering and transcriptomic profiles between control and treated condition indicate no effect of conditioned medium on immune cell populations. **E)** Summary of GSEA pathway analyses from distinct inflammatory pathways that was found significantly dysregulated in cluster I of the T cells in the first spheroid experiment.

8. The exclusion of the NF κ B pathway from the mechanism behind TAK1 was demonstrated by the treatment of an IKK inhibitor (TPCA-1) *in vitro*. However, this exclusion was not validated *in vivo*. To strengthen this assertion, it would be prudent to validate the exclusion of the NF κ B pathway *in vivo* as well, as the current exclusion may be considered somewhat premature without *in vivo* validation.

This comment is similar to point 1 raised by reviewer 2. In order to investigate this important aspect in the best possible way under physiological conditions, we were able to assess the effect of NF- κ B inhibition *in vivo* by gaining access to a dataset of a previously performed experiment (PMID: 27454298) in which the NF- κ B subunit RelA/p65 was deleted in the same KRAS^{G12D} mice (see **data set 17**). Importantly and in clear contrast to the TAK1 deficiency, the additional deletion of *Rela* did not prevent transdifferentiation and PDAC formation in KRAS^{G12D} mice. A careful pathological evaluation even indicated an exacerbation of PDAC development upon *Rela* ablation. This opposing effect of acinar cell-specific TAK1 vs. RelA deficiency further supports our conclusion that TAK1 mediates pancreatic oncogenesis through a function independent of activating NF- κ B signalling pathway.

Data set 17 / new Supplementary Fig. 2l, m: RelA is not necessary for $KRAS^{G12D}$ dependent PDAC development. **a** Haematoxylin and eosin (H&E) staining in pancreatic tissue sections from the indicated $KRAS^{G12D}$ and $KRAS^{G12D} RelA^{\Delta Ac}$ mice aged 46 to 61 weeks. **b** Quantification of healthy pancreas tissue (Normal), acinar-to-ductal metaplasia (ADM), pancreatic intraepithelial neoplasia 1 (PanIN-1) and PanIN-2 in the indicated animals ($n = 5$ for $KRAS^{G12D}$ mice and $n = 8$ for $KRAS^{G12D} RelA^{\Delta Ac}$ mice). The $KRAS^{G12D}$ mice share the same genetic background.

9. Figure 5g is not mentioned in the text, while Figure 5h is referenced in the text but appears to be absent. It is advisable to clarify whether Figure 5g was mistakenly referenced as Figure 5h, or if there was an oversight in the presentation of these figures.

We thank the reviewer for pointing out this mistake. All text references to the Figures are now correctly assigned.

Point-by-Point Response to the Referees:

We would like to thank all the reviewers for their positive evaluation of our revised manuscript.

The reviewers had no further questions.